# Learning Partial Concept Classes and Universal Rates Under Massart Noise

Ariel Avital [* 1]   Klim Efremenko [1]   Steve Hanneke [* 2]

## Abstract

The Massart noise condition is a central model in Probably Approximately Correct (PAC) learning theory. Its importance lies in it being an interpolation condition between realizable and the agnostic settings, under which one can attain faster rates than in the latter, and, under strict conditions, recover the rates of the former. Despite its importance, the Massart condition has not yet been fully explored in emerging extensions of statistical learning theory beyond the classical PAC framework. In this work, we present two such extensions. First, we revisit the transductive empirical risk minimization (TERM) algorithm of (Hanneke & Moran, 2026), and derive sharper excess error bounds under Massart noise using offset Rademacher techniques and local metric entropy introduced by (Zhivotovskiy & Hanneke, 2018). We then leverage this analysis to obtain new sample complexity bounds for PAC learning with partial concept classes and complete the characterization of universal rates under Massart noise.

## 1. Introduction

Probably Approximately Correct (PAC) learning, pioneered by (Valiant, 1984; Vapnik & Chervonenkis, 1974), has long served as a foundational framework in statistical learning theory. This framework is focused on the question of characterizing the best possible uniform learning rates —that is, error rates that can be guaranteed across all distributions. However, its minimax perspective and its inherent limitation in incorporating data-dependent assumptions have limited its applicability to real-world scenarios. More specifically,

by concentrating on worst-case distributions, PAC theory fails to capture optimal rates under a single, fixed data distribution, which is the typical scenario in most practical machine learning tasks (Cohn et al., 1994; Cohn & Tesauro, 1990). Furthermore, classical distribution-free PAC theory does not naturally incorporate data-dependent structural assumptions common in many learning problems, such as margin conditions or low-dimensional structure, directly into the concept class itself. While such settings have been extensively studied within statistical learning theory through distribution-dependent analyses and complexity-based techniques (Shawe-Taylor et al., 1998; Herbrich et al., 2000), these approaches generally rely on assumptions external to the combinatorial structure of the hypothesis class and therefore do not provide a purely distribution-free PAC characterization of learnability. These limitations have steered the development of alternative frameworks under statistical learning theory. Universal learning, introduced by (Bousquet et al., 2021), addresses the fixed-distribution scenario by exploring universal rates —the optimal rates achievable for any single, fixed distribution. Universal learning has revealed a richer landscape of rates and has been successfully applied across various problems, including multiclass, active, and interactive learning (Kalavasis et al., 2022; Hanneke et al., 2023; Hanneke & Xu, 2025; 2024; Hanneke et al., 2024; 2022). Another new line of research considers the incorporation of partial concept classes into the PAC theory. A partial concept class aims at modeling data-dependent assumptions by extending to concepts that are not necessarily defined over the entire instance space (Long, 2001; Alon et al., 2021). Among the central research questions arising from this extension, the determination of optimal sample complexity bounds has received significant attention Alon et al. (2021); Long (2001); Aden-Ali et al. (2023); Hopkins et al. (2022). Despite their extensive investigation into the realizable and agnostic settings, neither Universal Learning nor partial concept classes has yet been extended to binary classification under the Massart noise condition. The Massart noise (or $\beta$-bounded noise) condition is a critical setting where the Bayes classifier conditional probability of error is bounded by a known probability of $\beta < 1/2$, while the concept class is assumed to be realizable with respect to the Bayes classifier. Considered an intermediate noise model, it has been extensively studied under the standard PAC model (Massart & Nédélec, 2006; Giné & Koltchinskii,

*Equal contribution [1]Department of Computer Science, Ben-Gurion University, Be'er Sheva, Israel [2]Department of Computer Science, Purdue University, West Lafayette, Indiana, USA. Correspondence to: Ariel Avital <kish03@gmail.com>, Klim Efremenko <klim@bgu.ac.il>, Steve Hanneke <steve.hanneke@gmail.com>.

*Proceedings of the $43^{rd}$ International Conference on Machine Learning*, Seoul, South Korea. PMLR 306, 2026. Copyright 2026 by the author(s).

2006; Hanneke & Yang, 2015; Hanneke, 2016; Zhivotovskiy & Hanneke, 2018). To motivate our interest further, the gap between the realizable and agnostic settings varies, or yet, is unknown, under the aforementioned frameworks described above: (1) Realizable universal rates extend the known uniform rates by the faster exponential rate, while the agnostic universal rates fail to match the uniform rates, attaining exponential universal rate only for finite concept classes. (2) Sample complexity of partial concept classes has been shown to match classical PAC bounds under the realizable case Aden-Ali et al. (2023), yet current results cannot show the same for its agnostic counterpart. Due to this unpredictable behavior, the potential rates under Massart noise within these frameworks remain uncertain. In this work, we provide a complete investigation of universal rates and improved PAC bounds for partial concept classes under the Massart noise condition. Our core contribution is three-fold:

1. We sharpen the excess error bound of the transductive empirical risk minimization (TERM) algorithm variant of Hanneke & Moran (2026) under the Massart noise setting.

2. We use this algorithm to derive tighter sample complexity bounds for partial concept classes.

3. We provide a full characterization of the universal rates under bounded noise.

We emphasize that the main novelty of this work is analytical rather than algorithmic. The TERM procedure and the pattern-avoidance framework build on prior work in transductive and universal learning. Our contribution is to show that these tools can be made to operate in the intermediate Massart-noise regime, where neither the realizable nor the fully agnostic analyses apply directly. Technically, this requires a new excess-error analysis of TERM based on offset Rademacher complexity and local metric entropy, which then yields improved PAC bounds for partial concept classes and the missing Massart-noise cases in the universal-rate classification.

### 1.1. Background and Preliminaries

Following the classical supervised learning setting of binary classification, we consider an instance space $\mathcal{X}$, a concept class $\mathcal{H} \subseteq \{0,1\}^{\mathcal{X}}$ and a probability distribution $P$ over $\mathcal{X} \times \{0,1\}$. Given a classifier $h : \mathcal{X} \to \{0,1\}$, we define the error rate of $h$ to be $er_P(h) := \mathbf{P}[h(X) \neq Y]$, where $(X, Y) \sim P$. Our work focuses on distributions that satisfy the Massart noise condition, where its definition relies on the Bayes optimal classifier, $h_P^\star$, the minimizer of the classification error:

$$h_P^\star(x) := \mathbb{1}_{\{\mathbf{P}[Y=1|X=x] \geq 1/2\}}$$

As our work also involves learning partial concept classes, we will introduce two variants of $\beta$-bounded (Massart) noise assumptions: the strong $\beta$-bounded condition, which requires that the Bayes classifier $h_P^\star$ can be approximated by hypotheses in $\mathcal{H}$ in the measure sense, and the weak $\beta$-bounded condition, which only requires that the concept class $H$ will mimic $h_P^\star$ on every finite sample. We start with the former and postpone the definition of the latter to the presentation of partial concept classes.

**Definition 1** (Strong $\beta$-bounded (Massart) noise condition)**.** *A distribution $P$ over $\mathcal{X} \times \{0,1\}$ is said to satisfy $\beta$-bounded noise condition with respect to a concept class $\mathcal{H}$ if there exists $\beta \in (0, 1/2)$ such that each of the following holds almost surely:*

- $\mathbf{P}[h_P^\star(X) \neq Y|X] \leq \beta$

- $\inf_{h \in \mathcal{H}} \mathbf{P}[h(X) \neq h_P^\star(X)] = 0$

In what follows we abbreviate "strong/weak $\beta$-bounded noise condition" as "$\beta$-bounded" whenever the particular variant is evident from the context. To proceed with our presentation, a learner is given a set of $n$ i.i.d. samples from a $\beta$-bounded distribution $P$, with the objective of learning a classifier $\hat{h}_n : \mathcal{X} \to \{0,1\}$ that minimizes the error rate $er(\hat{h}_n)$. Unlike the realizable case, where the concept class can achieve arbitrarily small error, this guarantee does not hold under $\beta$-bounded noise. Hence, our focus shifts to the minimal excess error relative to the Bayes classifier, as formalized in the following definition.

**Definition 2** (Excess error)**.** *Let $\mathcal{H}$ be a concept class, and let $\left\{\hat{h}_n\right\}_{n \in \mathbb{N}}$ be the output of a learning algorithm. For any $\beta$-bounded distribution $P$ over $\mathcal{X} \times \{0,1\}$ and a sample $S_n := \{(X_i, Y_i)\}_{i \in [n]} \sim P^n$, we define its (expected) excess error as*

$$\mathcal{E}(n, P) := \mathbb{E}\left[er_P(\hat{h}_n) - er_P(h_P^\star)\right]$$

It is worth noting that our definition of excess error is taken with respect to the expected Bayes error, whereas the literature more commonly defines it relative to $\inf_{h \in \mathcal{H}} er_P(h)$. Moreover, Alon et al. (2021) argued that excess error should instead be measured with respect to the approximation error of the concept class, $er_P(\mathcal{H}) = \lim_{n \to \infty} \mathbb{E}[\min_{h \in \mathcal{H}} \hat{er}_{S_n}(h)]$, that is, the limiting expected empirical error of the concept class. This is a stricter notion that extends naturally to generalized settings, such as partial concept classes. We emphasize that Definition 2 coincides with both quantities: under the strong $\beta$-bounded noise condition, it is equivalent to the former, while under the weak $\beta$-bounded condition, it aligns with the latter under two sub-cases: Distributions with countable support, or concept classes with finite VC dimension, which is sufficient to show

that our sample complexity is a direct improvement over that of the agnostic sample complexity derived by (Alon et al., 2021). We derive these equivalences in Appendix B. Under both $\beta$-bounded noise conditions, PAC learning is characterized by two complexity measures of the concept class: the VC dimension and the star number. The formal definitions of the VC dimension and the star number are deferred to Appendix A. We denote the star number of a concept class $\mathcal{H}$ as $\mathfrak{s}(\mathcal{H})$. It is important to observe that a bounded star number ensures PAC learnability under $\beta$-bounded noise assumptions; this follows from the fact that a finite star number is a strictly stronger condition than a finite VC dimension.

**Partial Concept Classes** Unlike total concept classes— the standard objects of study in the PAC framework — partial concept classes may contain concepts that are not defined on certain regions of the instance space. To distinguish notation, we denote total concept classes by $\mathcal{H}$ and partial concept classes by $H$. The formal definition follows.

**Definition 3** (Partial concept class). *A partial concept class $H$ is a set of functions $h : \mathcal{X} \rightarrow \{0, 1, \star\}$, where $\star$ denotes an undefined value. The support of $h$ is defined as*

$$\mathrm{supp}(h) := \{x \in \mathcal{X} \mid h(x) \in \{0, 1\}\}.$$

Note that any prediction of $\star$ is formally treated as a prediction mistake. While all classical complexity measures -such as VC dimension and star number - extend naturally to partial concept classes, other definitions, such as distributional assumptions and PAC learnability must be generalized to capture the richer learning scenarios introduced by this extension. In the next definition, we generalize the $\beta$-bounded noise condition for partial concept classes. Before we state this condition, we denote $P_\mathcal{X}$ as the marginal distribution of $P$ over $\mathcal{X}$.

**Definition 4** (Weak $\beta$-bounded (Massart) noise condition). *A distribution $P$ over $\mathcal{X} \times \{0, 1\}$ is said to satisfy the generalized $\beta$-bounded noise assumption with respect to a partial concept class $H$ if there exists $\beta \in [0, 1/2)$ such that each of the following holds almost surely:*

- $\mathbf{P}\left[h_P^\star(X) \neq Y | X\right] \leq \beta$

- *For every $n \in \mathbb{N}$ and $X_1, \ldots, X_n \sim P_\mathcal{X}^n$ there exists $h \in H$ such that $h(X_i) = h_P^\star(X_i)$ holds for all $i \in [n]$.*

Indeed, we can observe that Definition 4 is weaker than Definition 1 in the sense that the concept class can only mimic $h_P^\star$ over every finite random set, yet there may be scenarios where the condition $\inf_{h \in \mathcal{H}} \mathbf{P}\left[h(X) \neq h_P^\star(X)\right] = 0$ will not hold[1].

---

[1] For a counterexample on the realizable setting, see (Alon et al., 2021)[Lemma 33].

**Measurability Assumptions and Additional Notation** Universal learning algorithms are based on the notion of Gale-Stewart games (Gale & Stewart, 1953), and thus require certain measurability assumptions to ensure their well-definedness. To this end, we assume $\mathcal{X}$ to be a Polish space and our (total) concept class $\mathcal{H}$ to be measurable, that is, that $\mathcal{H}$ could be parameterized with respect to some Polish space $\Theta$. This assumption, which is known as *the image admissible Souslin* property and is a standard condition in empirical process theory, ensures the universally measurability of our learning algorithm[2]. We make frequent use of conditional probabilities and expectations. Unless stated otherwise, all such quantities are understood in the standard measure-theoretic sense, with statements holding almost surely with respect to an appropriate version.

We finish this section with the following conventions and definitions. Every logarithm is considered a natural logarithm, unless stated otherwise, e.g. $\log_2(x)$. We write $Log(x)$ for $\max\{1, \log(x)\}$. We denote $\hat{er}_S(h)$ as the empirical error of $h$ on the sample $S \subseteq \{\mathcal{X} \times \{0, 1\}\}^*$, that is, $\hat{er}_S(h) := \sum_{(x,y) \in S} \mathbb{1}_{\{h(x) \neq y\}}$. We may sometimes wish to consider the error rate conditioned over a specific event $A$. To this end, we define $er_{|A}(h) := \mathbf{P}\left[h(X) \neq Y | A\right]$. We define $\mathcal{X}_S := \{x \in \mathcal{X} : (x, \cdot) \in S\}$ to be the set of distinct points $x \in \mathcal{X}$ of some sample $S \in \{\mathcal{X} \times \{0, 1\}\}^*$.

## 1.2. Related Work

**Massart Noise** Early PAC learning research focused on two primary regimes: the realizable setting, where the (total) concept class is assumed to contain the target concept, achieving error rates of $\frac{\mathrm{VC}(\mathcal{H})}{n}$, and the agnostic setting, where no distributional assumptions yield rates of $\sqrt{\frac{\mathrm{VC}(\mathcal{H})}{n}}$. The first work to present intermediate rates in the context of PAC learning was that of (Tsybakov, 2004). These rates were shown to be possible when the target distribution exhibits a margin type condition, also known as Bernstein class condition, which has its roots in discriminative analysis (Mammen & Tsybakov, 1999). A series of successive works refined their results (Massart & Nédélec, 2006; Giné & Koltchinskii, 2006; Hanneke & Yang, 2015) by introducing new quantities such as the disagreement coefficient and the star number. The $\beta$-bounded noise condition is often named after (Massart & Nédélec, 2006), who noticed it as a special interpretable case of the Bernstein class condition. Current PAC sample complexity estimates for Massart noise condition contain the upper bound $\frac{\mathrm{VC}(H)Log\left(\mathfrak{s}(H) \wedge \frac{(1-2\beta)^2}{d\varepsilon}\right) + Log\left(\frac{1}{\delta}\right)}{\varepsilon(1-2\beta)}$ of (Hanneke & Yang, 2015), and the lower bound

---

[2] We emphasize that our algorithms rely on universally measurable functions introduced by Bousquet et al. (2021); Hanneke et al. (2024), see pointers therein for more details.

$\frac{\text{VC}(H)+\beta\text{Log}\left(\mathfrak{s}(H)\wedge\frac{(1-2\beta)^2}{\varepsilon}\right)+\text{Log}\left(\frac{1}{\delta}\right)}{\varepsilon(1-2\beta)}$ of (Hanneke, 2016), which builds upon the information-theoretic techniques of (Raginsky & Rakhlin, 2011). (Zhivotovskiy & Hanneke, 2018) have suggested an alternate complexity term to characterize empirical risk minimization (ERM) learnability, which they termed the *fixed point of a local empirical entropy*. Through this quantity, they were able to recover the aforementioned sample complexity, as it was shown to be equivalent to a function of the VC dimension and the star number.

**Partial Concept Classes**  Classical PAC theory focuses exclusively on total concept classes. Early appearances of $\{0, 1, \star\}$-valued functions arose mainly as technical tools to extend generalization bounds, such as in Sauer's lemma (Haussler & Long, 1995) and the analysis of [0,1]-valued function classes (Bartlett & Long, 1998). A direct study of partial concept classes was initiated by Long (2001), who established their agnostic PAC learnability. After a long hiatus, Alon et al. (2021) revisited the problem and introduced several important results and insights. Most notably, they identified key challenges unique to partial concept classes—demonstrating the failure of empirical risk minimization (ERM) and the absence of any natural learning algorithm in this setting. They further showed that disambiguating a partial class (i.e., converting it into a total class by replacing $\star$ with $\{0, 1\}$) results in a total class of size $|\mathcal{X}|^{O(\text{VC}(H)\log|\mathcal{X}|)}$, incurring substantial complexity inflation. Despite these difficulties, Alon et al. (2021) proposed transductive algorithms that improved sample-complexity bounds for both realizable and agnostic cases, leveraging techniques ranging from sample compression to boosting of the 1-Inclusion Graph algorithm (Haussler et al., 1994). However, their results did not fully close the sample-complexity gap in either setting. Subsequently, Aden-Ali et al. (2023) achieved realizable sample-complexity bounds matching those of classical PAC learning, using martingale inequalities with the 1-Inclusion Graph algorithm, while Hopkins et al. (2022) derived agnostic rates via an agnostic-to-realizable reduction, though their bounds remain looser than those in Alon et al. (2021). It is important to note that the sample complexity derived in the works of Long (2001) and Hopkins et al. (2022) is expressed in excess error with respect to $\inf_{h\in H} er_P(h)$, which can be loose in the case of partial concept classes [3].

**Universal Learning**  Universal learning is a relatively new framework in the landscape of statistical learning theory, using ideas and tools from descriptive set theory, online learning and game theory. This line of work, which was initiated by (Bousquet et al., 2021), aimed at characterizing

the expected error of a learning algorithm over any fixed distribution, rather than trying to catch the worst case behavior over all distributions at once, that is, trying to catch the individual "learning curves" rather than the collective outer envelope. Starting under the most basic setting of binary classification under realizable distribution, (Bousquet et al., 2021) revealed a trichotomy of universal rates: exponential, linear or arbitrarily slow. While (Bousquet et al., 2021) claimed that the constants for universal rates are distribution-dependent, (Bousquet et al., 2023) proved that the linear rate is in fact distribution-independent, and is characterized by the VCL dimension. A follow-up work of (Hanneke & Moran, 2026) gave full characterization of the agnostic case, with the surprising tetrachotomy of $e^{-n}, e^{-o(n)}, o\left(n^{-1/2}\right)$ or arbitrarily slow. Among the multiple extensions made to other learning problems, the closest are those that characterize universal rates for empirical risk minimizers for both the realizable (Hanneke & Xu, 2024) and the agnostic cases (Hanneke & Xu, 2025). Their results share similarity to general universal rates: The realizable rates extend the uniform known rates by the exponential rate to obtain the tetrachotomy of rates: $e^{-n}, 1/n, \log(n)/n$ and arbitrarily slow, while the agnostic rate has the trichotomy of $e^{-n}, o\left(n^{-1/2}\right)$ and arbitrarily slow. However, in contrast to our setting, the characterization of ERM universal rates relies on different complexity measures: The star-eluder sequence and the VC dimension. Another notable work characterizes the universal rates for active learning (Hanneke et al., 2024); we employ their star-set elimination algorithm as a critical step in establishing the universal linear rate.

## 2. Main Results

**Learning Partial Concept Classes under Massart Noise** The main caveat of learning partial concept classes stems from the fact that one could not rely on empirical risk minimization. Indeed, a concept with a small empirical error on a labeled data set gives no indication regarding its population error, as it may be undefined on the majority of the distribution's support. However, under the assumption that almost surely any projection of the concept class on a finite data set contains a fully-defined concept, one may succeed in making local predictions on observed unlabeled data points using transductive learning algorithms. Recently, (Hanneke & Moran, 2026) have applied a variant of the TERM algorithm by (Vapnik & Chervonenkis, 1974) for learning partial concept classes under Bayes-realizable distributions[4], as a part of their agnostic universal characterization. This algorithm was shown by (Hanneke & Moran, 2026) to have an

---

[3]See Appendix C in Alon et al. (2021) for further discussion.

[4]In the context of partial concept class, this means that for any random finite sample, there exists a concept that is fully defined on all points in the sample with probability one. See (Hanneke & Moran, 2026) for more details.

in-expectation bound of $o\left(\sqrt{\frac{\mathrm{VC}(\mathcal{H})}{n}}\right)$. We sharpen this result for $\beta$-bounded distributions using offset Rademacher complexity and local metric entropy as were employed by (Zhivotovskiy & Hanneke, 2018). Before stating the main result, we let $\mathbb{A} : H \times (\mathcal{X} \times \{0,1\})^n \times \mathcal{X} \to \{0,1\}$ denote the prediction rule of the TERM algorithm, given a partial concept class $H$, a labeled sample of $n$ points, and a query point in $\mathcal{X}$.

**Theorem 1.** *Let* $S = \{(X_1, Y_1), ..., (X_n, Y_n)\} \sim P^n$, *where* $P$ *satisfies Definition 4 with respect to* $H$. *Let* $\hat{h}_n(\cdot) := \mathbb{A}(H, S, \cdot)$, *then the following hold*

$$\mathcal{E}(n, P) = O\left(\frac{d \log\left(\min\left\{\frac{n(1-2\beta)}{d}, \frac{\mathfrak{s}}{1-2\beta}\right\}\right)}{(1-2\beta)n}\right)$$

*where* $d$ *and* $\mathfrak{s}$ *are the VC dimension and star number of* $H$, *respectively.*

**Sample Complexity for Learning Partial Concept Classes Under Massart Noise** While significant progress has been made in characterizing the sample complexity for partial concept classes in both the realizable and agnostic settings, the case of Massart noise has yet to be addressed. Consequently, the best currently known estimates are those of Alon et al. (2021) for the agnostic setting: $O\left(\frac{\mathrm{VC}(H)\log^2(d/\varepsilon) + \log(1/\delta)}{\varepsilon^2}\right)$ and $\Omega\left(\frac{\mathrm{VC}(H) + \log(1/\delta)}{\varepsilon^2}\right)$. Beyond the looseness that persists in the upper bound, these estimates are significantly looser than the corresponding sample complexity under the classical PAC model with Massart noise. By employing the TERM algorithm in conjunction with confidence boosting, we establish tighter bounds, as stated in the following theorem.

**Theorem 2.** *For any partial concept class $H$, let $\mathcal{M}(\varepsilon, \delta)$ denote the sample complexity for PAC learning $H$ under weak $\beta$-bounded distributions (Definition 4). Then $\mathcal{M}(\varepsilon, \delta)$ satisfies the following bounds:*

- $\mathcal{M}(\varepsilon, \delta) = O\left(\frac{d}{\varepsilon(1-2\beta)} \log\left(\min\left\{\frac{1}{\varepsilon}, \frac{\mathfrak{s}}{1-2\beta}\right\}\right) \log\left(\frac{1}{\delta}\right)\right)$

- $\mathcal{M}(\varepsilon, \delta) = \Omega\left(\frac{d + \log\left(\frac{1}{\delta}\right) + \beta \log\left(\min\left\{\frac{1}{\varepsilon}, \mathfrak{s}\right\}\right)}{\varepsilon(1-2\beta)}\right)$

*where* $d$ *and* $\mathfrak{s}$ *are the VC dimension and star number of* $H$, *respectively.*

*Remark* 2.1. The excess-error notion considered here coincides with the stricter approximation-error mentioned in the previous section for every PAC learnable class, that is, any concept class with finite VC dimension. Hence, its operational interpretation holds also with respect to the latter.

**Universal Rates under Massart Noise** We proceed to our second application of the TERM algorithm, which completes the characterization of universal learning rates under Massart noise. It is important to note that this analysis pertains to total concept classes and assumes distributions that satisfy the strong $\beta$-bounded noise condition (Definition 1). Before presenting our results, we first define universally learnability under Massart noise (Definition 1) by extending the definition used by (Bousquet et al., 2021).

**Definition 5.** *Let $\mathcal{H}$ be a (total) concept class and $R(n) \to 0$ be a rate function. We say that:*

- $\mathcal{H}$ *is $\beta$-universally learnable at rate $R$ if there exists a learning algorithm $\hat{h}_n$ such that for every $\beta$-bounded distribution $P$, there exist $C, c > 0$ for which its excess error satisfies $\mathcal{E}(n, P) \leq C R(cn)$ for all $n$.*

- $\mathcal{H}$ *is not $\beta$-universally learnable at rate faster than $R$ if for every learning algorithm $\hat{h}_n$, there exists a $\beta$-bounded distribution $P$ and $C, c > 0$ for which its excess error satisfies $\mathcal{E}(n, P) \geq C R(cn)$ for infinitely many $n$.*

- $\mathcal{H}$ *is $\beta$-universally learnable with optimal rate $R$ if $\mathcal{H}$ is learnable at rate $R$ and $\mathcal{H}$ is not learnable faster than $R$.*

- $\mathcal{H}$ *requires arbitrarily slow rates if, for every $R(n) \to 0$, $\mathcal{H}$ is not learnable faster than $R$.*

When the learning rate is stated with an asymptotic notation rather than with an exact function, it requires a more rigorous treatment. For this matter, we adapt the following definition from Hanneke & Moran (2026).

**Definition 6.** *Following the notations in Definition 5, we say that $\mathcal{H}$ is $\beta$-universally learnable with optimal rate exactly $e^{-o(n)}$, if for every $T(n) = o(n)$, $\mathcal{H}$ is $\beta$-universally learnable at rate $R(n) = e^{-T(n)}$ and for every learning algorithm $\hat{h}_n$, there exists a function $T(n) = o(n)$ and a $\beta$-bounded distribution $P$ such that $\mathcal{E}(n, P) \geq e^{-T(n)}$ for infinitely many $n$.*

Since the Massart noise model essentially interpolates between the realizable and agnostic settings, several parts of its universal characterization are direct consequences of existing results.

- **If $|\mathcal{H}| < \infty$, then $\mathcal{H}$ is $\beta$-universally learnable with an optimal rate $e^{-n}$.** The fact that no learning algorithm can learn at a rate faster than exponential under realizable distribution is originally due to (Schuurmans, 1997). A matching lower bound for the agnostic case follows from the elementary lower bound on the sample complexity of determining the polarity of a slightly

biased coin (Hanneke & Moran, 2026)[5]. The upper bound is trivially obtained using an ERM learner, by applying Chernoff inequality with the union bound (Hanneke & Moran, 2026).

- **If $|\mathcal{H}| = \infty$, but does not have an infinite Littlestone tree, then $\mathcal{H}$ is $\beta$-universally learnable with optimal rate $e^{-o(n)}$.** The upper bound was proved in agnostic setting by (Hanneke & Moran, 2026). Their use of a $\beta$-bounded distribution in their construction of the lower bound imply that no further improvement can be applied in our setting.

- **If $\mathcal{H}$ has an infinite Littlestone tree, then $\mathcal{H}$ is not $\beta$-universally learnable at a rate faster than $1/n$. Furthermore, if $\mathcal{H}$ has an infinite VCL tree, then $\mathcal{H}$ requires arbitrarily slow rates.** While (Bousquet et al., 2021) established these results within the realizable setting, the conclusion naturally extends to the more general—and strictly more challenging—regime considered in our work.

To this end, we complete the characterization establishing two additional universal rates which, interestingly, coincide with the known uniform rates under the same distributional assumption, thus reaching a pentachotomy of rates. We defer the complexity measures associated with the next theorem to Appendix E.

**Theorem 3.** *For every concept class $\mathcal{H}$ with $|\mathcal{H}| \geq 2$, the following hold:*

- *If $|\mathcal{H}| < \infty$, then $\mathcal{H}$ is $\beta$-universally learnable with an optimal rate exactly $e^{-n}$.*

- *If $|\mathcal{H}| = \infty$, but does not have an infinite Littlestone tree, then $\mathcal{H}$ is $\beta$-universally learnable with optimal rate exactly $e^{-o(n)}$.*

- *If $\mathcal{H}$ has an infinite Littlestone tree but does not have an infinite star tree, then $\mathcal{H}$ is $\beta$-universally learnable with optimal rate exactly $\frac{1}{n}$.*

- *If $\mathcal{H}$ has an infinite star tree but does not have an infinite VCL tree, then $\mathcal{H}$ is $\beta$-universally learnable with optimal rate exactly $\frac{\log(n)}{n}$.*

- *If $\mathcal{H}$ has an infinite VCL tree, then $\mathcal{H}$ requires arbitrarily slow rates.*

[5]An extension has to be made for the case of $H = \{h, 1 - h\}$, where one sample is sufficient if the distribution is realizable.

## 3. Transductive Learning Algorithm and Sample Complexity for Partial Concept Classes

In this section, we present the transductive empirical risk minimization (TERM) algorithm of (Hanneke & Moran, 2026) and give a brief account of our proof of its excess error bound. The full proof can be found in Appendix C. Given a labeled data set of size $n$ and an additional *unlabeled* point, the algorithm outputs the prediction among all the total projection that achieves the smallest empirical error on the *first half* of the sample, where by total projections we mean all the projections of the concept class that are defined on both the data set and the test point. More formally, denote $a_{\leq k} := \{a_1, a_2, \ldots, a_k\}$ for some sequence $a$ and let $\text{TERM}_n^H : (\mathcal{X} \times \{0,1\})^{\lceil n/2 \rceil} \times \mathcal{X}^{\lfloor n/2 \rfloor + 1} \to \{0,1\}^{n+1}$ be the universally measurable mapping [6] defined as

$$\text{TERM}_n^H \left( x_{\leq \lceil n/2 \rceil}, y_{\leq \lceil n/2 \rceil}, x_{\lceil n/2 \rceil + 1}, \ldots, x_{n+1} \right) := \underset{h \in H[x_1, \ldots, x_{n+1}]}{argmin} \hat{er}_{S_{\leq \lceil n/2 \rceil}}(h),$$

then given a sample $S = (x_1, y_1, \ldots, x_n, y_n)$ and a test point $x_{n+1}$, our algorithm returns the prediction

$$\mathbb{A}\left(H, S, x_{n+1}\right) :=$$
$$\text{TERM}_n^H \left( x_{\leq \lceil n/2 \rceil}, y_{\leq \lceil n/2 \rceil}, x_{\lceil n/2 \rceil + 1}, \ldots, x_{n+1} \right)(x_{n+1}).$$

This approach allows one to use the remaining half of the sample to transform our analysis from expected excess error to expected excess *empirical* error using exchangeability arguments. That is, let $S$ be a random i.i.d sample sequence, and $S_{i:j}$ be its subset containing the indices $i$ to $j$, one can argue that

$$\mathcal{E}(n, P) = \mathbb{E}\left[\hat{er}_{S_{\lceil n/2 \rceil + 1 : n + 1}}(\hat{h}_n) - \hat{er}_{S_{\lceil n/2 \rceil + 1 : n + 1}}(h_P^\star)\right].$$

From this point on, (Hanneke & Moran, 2026) proceed by separating the analysis of labels from that of the sample covariates, and then apply empirical variants of concentration inequalities based on localization arguments. Although this approach yields tight error bounds in the agnostic case, it remains insufficient to address more flexible noise regimes, such as Massart noise. To attend to this issue, we turn to another localization-based analysis that is known to be more subtle in these settings, namely the *fixed point of local entropy* by (Zhivotovskiy & Hanneke, 2018). Using symmetrization and contraction, this technique reduces the analysis of the ERM excess error to obtaining a bound over the shifted empirical centered sub-gaussian processes defined over the functional classes, which are based on the concept class. Finally, (Zhivotovskiy & Hanneke, 2018) show that such quantities can be controlled by the fixed

[6]For more details, see (Hanneke & Moran, 2026).

quantity of a local empirical entropy, a quantity equivalent (under regularity conditions) to the complexity term associated with learning Massart noise. However, the application of this technique is not straightforward, as we seek to bound the conditional excess empirical error rather than the excess error itself. In addition, the derivation of (Zhivotovskiy & Hanneke, 2018) considers only total concept classes, while we deal with partial concept classes. To overcome these, we apply a symmetrization trick over the excess empirical error and apply a contraction lemma that allows us to drop the labels and focus on local structure of the concept class. We then use the fact that the analysis involves local empirical entropy and restrict our attention to the *total* version space over the sample. As this set is guaranteed to be non-empty due to the Massart condition, we can apply the empirical entropy bound as-is, bounding the version space complexity measures it inherits from the original (partial) concept class.

We finish this section with a brief sketch of our upper sample complexity derivation. Our estimate is derived by applying *Boosting*, a standard method in statistical machine learning that allows the conversion of an in-expectation algorithm into a high-probability learner. The full proof of Theorem 2, including the lower bound, is deferred to Appendix D. The sample is split into two parts. The first part is also split into equally sized batches, where the batch size is chosen such that a learner trained on each of these batches has excess risk more than $\varepsilon/2$ with probability at most $1/2$. We set the number of batches to be $\lceil log_2(2/\delta) \rceil$, to ensure that the probability of the event that all of the classifiers have excess risk error greater than $\varepsilon/2$ is at most $\delta/2$. The second part of the sample is then used to pick a sufficiently good classifier based on the empirical error of each of the $\lceil log_2(2/\delta) \rceil$ classifiers. Using union bound together with Bernstein and Chernoff inequalities, we can guarantee that a classifier with excess risk greater than $\varepsilon/2$ is chosen, with probability at most $\delta/2$. The argument follows by an additional union bound.

## 4. Universal Rates

This section is dedicated to the remaining items needed to acquire full characterization of the universal rates under Massart noise. These are formulated in the next couple of theorems. Starting with the upper bounds, we have chosen to couple both upper bounds together in the next theorem, as they rely on the same technical proof, save the guarantee that the TERM algorithm can provide for either prediction using partial concept classes with finite star number or finite VC dimension.

**Theorem 4.** *Let $\mathcal{H}$ be a concept class, then:*

- *If $\mathcal{H}$ has an infinite Littlestone tree but does not have an infinite star tree, then $\mathcal{H}$ is $\beta$-universally learnable*

*at rate $\frac{1}{n}$.*

- *If $\mathcal{H}$ has an infinite star tree but does not have an infinite VCL tree, then $\mathcal{H}$ is $\beta$-universally learnable at rate $\frac{\log(n)}{n}$.*

The learning algorithm for Theorem 4 is depicted in Algorithm 1. Next, we move onto our lower bound, this result uses similar ideas from the lower bounds of (Hanneke, 2016; Raginsky & Rakhlin, 2011).

**Theorem 5.** *If $\mathcal{H}$ has an infinite star tree, then for any learning algorithm $\hat{h}_n$, there exists a $\beta$-bounded distribution $P$ (Definition 1) such that*

$$\mathcal{E}(n, P) \gtrsim \frac{\log n}{n}$$

*for infinitely many $n$.*

The remainder of this section outlines the ideas behind the upper bounds, the formal proofs of Theorems 4 and 5 can be found in Appendices E and F, respectively. We can observe a sense of parallelism between the VCL and star trees to the VC dimension and star number. Indeed, we will work towards transforming these universal complexity measures into their uniform counterparts. Importantly, for each such pair, boundedness of one measure does not imply boundedness of the other; see (Bousquet et al., 2021; Hanneke et al., 2024) for counterexamples. Our proof is similar to the universal $o\left(n^{-1/2}\right)$ upper bound of (Hanneke & Moran, 2026): We opt to distill a total concept class with universal complexity measure into multiple partial concept classes with uniform complexity measure that corresponds to the universal one, and circumvent the problem of noisy labels by enumerating all possible labeling combinations.

For simplicity, we draw our sketch for a concept class $\mathcal{H}$ that has no infinite star tree (the equivalent sketch for the case of no infinite VCL tree is identical). We start by fixing some $\beta$-bounded $P$ w.r.t to $\mathcal{H}$. Recall that Theorem 1 assures linear expected excess error for learning some finite star number partial concept class $H$, given that the data is sampled from a (weak) $\beta$-bounded distribution $P$ (w.r.t to $H$). It is possible to extend this result under more flexible, weaker conditions. Assume that we have $k$ independently *random*, finite star partial concept classes $\hat{H}^1, \ldots, \hat{H}^k$, where each has probability greater than $1/2$ to include $h_P^\star$ labeling on a sample $S = (X_1, Y_1, \ldots, X_m, Y_m)$ and a test point $X$ (the randomness of the partial concept classes is independent of $S, X$). It is not hard to see that a majority voting on the collective TERM prediction will restore the error bound: The probability that at least half of these partial concept classes does not contain $h_P^\star$ labeling on $\mathcal{X}_S \cup \{X\}$ is exponentially small (thus ensuring that $P$ is weakly $\beta$-bounded w.r.t to each such partial concept class that does contain $h_P^\star$ labeling on $\mathcal{X}_S \cup \{X\}$), whilst Markov's inequality recovers the same

**Algorithm 1** Learning Algorithm for Universal Learning using pattern avoidance functions

1: **Input:** Sample $S = \{(X_i, Y_i)\}_{i=1}^n$ of size $n$, TERM algorithm $\mathbb{A} : H \times (\mathcal{X} \times \{0,1\})^n \times \mathcal{X} \to \{0,1\}$ and a test point $X$.
2: **Output:** A prediction on $X$.
3: $m_1 \leftarrow \lfloor n/3 \rfloor$, $m_2 \leftarrow \lfloor 2n/3 \rfloor$
4: Set $S_1 := \{X_i\}_{i=1}^{m_1}$, $S_2 := \{(X_i, Y_i)\}_{i=m_1+1}^{m_2}$, $S_3 := \{(X_i, Y_i)\}_{i=m_2+1}^n$
5: **Part I - Learning pattern avoidance functions**
6: For every batch size $b \in [m_1]$, do the following:
7: a. Split $S_1$ into disjoint $b$-sized batches: $\{S_{1,i}^b \mid i \in [\lfloor m_1/b \rfloor]\}$
8: b. For every $i \in [\lfloor m_1/b \rfloor]$, learn the pattern avoidance function set $\hat{F}_b^i$ on $\mathcal{X}_{S_{1,i}^b}$ using every possible labeling.
9: c. Set $\tilde{H}_b^i := \bigcup_{f \in \hat{F}_b^i} \tilde{H}_b^{i,f}$, where $\tilde{H}_b^{i,f}$ denotes the partial concept class induced by $f$ on $\mathcal{X}_{S_2}$.
10: **Part II - Evaluation of sufficiently good batch size**
11: For every batch size $b \in [m_1]$, do the following:
12: a. Denote by $\tilde{H}_b^i(\mathcal{X}_{S_2})$ the set of classifications of $\mathcal{X}_{S_2}$ realizable by $\tilde{H}_b^i$.
13: b. Calculate $\hat{er}_{S_2}(\tilde{H}_b^i)$, the minimal empirical error on $S_2$ among classifications in $\tilde{H}_b^i(\mathcal{X}_{S_2})$.
14: c. Select $\hat{b}$ to be the smallest $b \in [m_1]$ such that at least $8/10$ of its error estimates $\left\{\hat{er}_{S_2}(\tilde{H}_b^i)\right\}_{i \in [\lfloor m_1/b \rfloor]}$ are at most $2\sqrt{\frac{b'2^{b'}\log(n^3)}{n}}$ larger compared to every error estimate in $\left\{\hat{er}_{S_2}(\tilde{H}_{b'}^{i'})\right\}_{i' \in [\lfloor m_1/b \rfloor]}$, where $b < b' \leq m_1$.
15: **Part III - Predicting TERM-based majority voting with the chosen pattern avoidance functions set**
16: a. Set $\hat{H}_{\hat{b}}^i := \bigcup_{f \in \hat{F}_{\hat{b}}^i} \hat{H}_{\hat{b}}^{i,f}$, where $\hat{H}_{\hat{b}}^{i,f}$ denotes the partial concept class induced by $f$ on $\mathcal{X}_{S_3} \cup \{X\}$.
17: b. Return the majority vote of the set $\left\{\mathbb{A}\left(\hat{H}_{\hat{b}}^i, S_3, X\right)\right\}_{i \in [\lfloor m_1/\hat{b} \rfloor]}$.

excess error as before, conditioned on the complementary event. Thus, it is sufficient for our cause to construct a finite set of these random partial concept classes.

Fortunately, the creation of such partial concept classes was shown to be possible by (Bousquet et al., 2021; Hanneke et al., 2024) using three key results: (1) The non-existence of an infinite tree is translated into the existence of a winning strategy in a Gale-Stewart game. (2) An algorithm based on such winning strategy is ensured to not observe "forbidden" data patterns (star sets in our case) on a consistent data sequence with the concept class. (3) Transforming this guarantee to the probabilistic setting for random realizable data sequences, we can show that the probability of observing these "forbidden" data patterns decays with the sequence length. These are summarized in the next lemma, the definition of realizable distribution can be found on Appendix A.

**Lemma 6** (Informal). *Let $\mathcal{H}$ be a concept class that does not have an infinite star tree, and let $X_1, Y_1, \ldots$ be an i.i.d sequence sampled from a **realizable** distribution w.r.t to $\mathcal{H}$. Then there exists a universally measurable pattern avoidance function $\tilde{g}_t$ that satisfies $\mathbf{P}\left[per(\tilde{g}_t) > 0\right] \to 0$, where $per(\tilde{g}_t)$ denotes the (random) probability that $\tilde{g}_t$ encounters star patterns.*

Given the above lemma, it is guaranteed that there exists a batch size $b^*$ such that $\mathbf{P}\left[per(\tilde{g}_t) > 0\right] \leq 1/10$, if the sequence is sampled from a 0-bounded distribution w.r.t to $\mathcal{H}$ (that is, a sample from a realizable distribution $P^*$ satisfying $P_\mathcal{X}^* = P_\mathcal{X}$ and $P^*\left[Y = h_P^*(X)|X\right] = 1$). Assuming that we know the value of $b^*$, we may easily create the desired partial concept classes, even if our data is sampled from a $\beta$-bounded distribution rather than a realizable one. Securing additional $k$ independent sample batches of size $b^*$, $S_{B_1}, \ldots, S_{B_k}$, we use each batch to learn a set of $2^{b^*}$ pattern avoidance functions $\left\{\hat{F}_{b^*}^i\right\}_{i \in [k]}$, one for every possible labeling of the batch. We then use $\hat{F}_{b^*}^i$ and an additional independent sample $S$ to construct $\hat{H}^i$ over $\mathcal{X}_S \cup \{X\}$, by including every labeling $\{0,1\}^{\mathcal{X}_S \cup \{X\}}$ such that none of the functions in $\hat{F}_{b^*}^i$ witness it to contain a forbidden star pattern. We claim that each $\hat{H}^i$ independently has a finite star number, and that at least $8/10$ of the partial concept classes contains $h_P^*$ labeling. Indeed, each partial concept class has a star number of at most $b^*2^{b^*}$, as it is a union of at most $2^{b^*}$ concept class with star number $b^*$. To prove the second claim, we observe that it is sufficient to show that $h_P^*$ labeling is not eliminated by at least one pattern avoidance function in the set $\hat{F}_{b^*}^i$: The function that is learned on $S_{B_i}$ using $h_P^*$ labeling. By Lemma 6, as each batch is of length at least $b^*$, the probability that this function observes a star pattern is at most $1/10$, the second claim follows by Hoeffding inequality.

This leaves us with the remaining obstacle of estimating the desired batch size $b^*$. For this task, we require an additional validation sample. From now on, we drop the previous sample conventions and instead partition the sample $S$ to three equal parts: $S_1$ is used to learn all possible pattern avoidance functions up to batch size $\lfloor n/3 \rfloor$, producing forbidden pattern functions sets $\left\{\hat{F}_b^i\right\}_{i \in [\lfloor n/3b \rfloor]}$ for every $b \in [\lfloor n/3 \rfloor]$. $S_2$ is then used for batch size estimation, by building partial concept classes based on each forbidden patterns functions set $\left\{\hat{F}_b^i\right\}$ over $\mathcal{X}_{S_2}$ and evaluating their empirical error on $S_2$. Finally, $S_3$ is used for the actual prediction using the TERM majority voting predictor. As we do not know the identity of $h_P^*$, we perform our estimation of $b^*$, denoted $\hat{b}$, in a relative manner: For sufficiently large $n$, the majority of a forbidden-pattern functions set $\left\{\hat{F}_b^i\right\}$ attain low validation error once the batch size $b$ exceeds an appropriate

threshold. Accordingly, we select our estimate $\hat{b}$ to be the smallest batch size for which most of its forbidden pattern functions perform as well as any pattern avoidance functions set based on a greater batch size than $\hat{b}$. More specifically, by conditioning on the support of $\mathcal{X}_{S_2}$, we use uniform convergence arguments to select $\hat{b}$ to be the infimum of the set of all batches sizes for which we are able to construct partial concept classes, such that at least $8/10$ of them has empirical error at most $2\sqrt{\frac{b' 2^{b'} \log(n^3)}{n}}$ larger than the empirical error of every partial concept class built by pattern avoidance function based on larger batch sizes $b'$. We show in Appendix E (Lemma 17) that this leads to batch size estimate $\hat{b}$ for which $\mathbf{P}\left[per(\tilde{g}_{\hat{b}})\right] \leq 3/10$ with probability of at least $1 - \frac{C}{n}$ for some constant $C > 0$, which is sufficient for our needs.

## 5. Conclusion

This paper has extended two separate learning theories to the case of Massart noise using a new excess error bound for the TERM algorithm. Our first application allowed us to derive new PAC sample complexity bounds for partial concept classes, while our second application allowed us to show the existence of linear and $\log(n)/n$ universal rates, allowing us to complete the picture of universal learning rates with a complementary lower bound. Although our sample complexity bounds are considerably tighter than previous estimates, standard PAC estimates suggest potential for further refinement. We conjecture that the transductive methods introduced by (Aden-Ali et al., 2023) and (Dughmi et al., 2025) can be extended to this regime. A further objective is the extension of universal rates to the Bernstein-class condition. As extremal rates are already determined, the central question is whether intermediate rates mirror the uniform rate behavior observed in Massart noise—an outcome consistent with our current conjectures.

## Acknowledgments

Funding: Klim Efremenko and Ariel Avital were supported by the European Research Council Grant number: 949707. Steve Hanneke acknowledges support by grant no. 2024243 from the United States - Israel Binational Science Foundation (BSF). The authors would also like to thank the anonymous reviewers for their valuable feedback on earlier versions of this manuscript.

## Impact Statement

This paper studies theoretical aspects of statistical learning under the Massart noise condition, with a focus on universal learning rates and partial concept classes. The work is primarily mathematical and foundational in nature, and does not introduce a deployable machine learning system or a new dataset. Its main contribution is the development of tighter learning guarantees and complexity characterizations for binary classification under structured noise assumptions.

Potential positive impacts of this work include improving the theoretical understanding of learning under realistic noise models, which may indirectly contribute to the design of more reliable and data-efficient machine learning algorithms. In particular, the results may help clarify the conditions under which fast learning rates are achievable, and may support future work on robust learning methods in settings involving incomplete or uncertain information.

Because the paper is theoretical, we do not anticipate direct harmful societal consequences arising specifically from the presented results. However, as with most advances in machine learning theory, the techniques developed here could eventually contribute to systems deployed in high-impact domains such as automated decision-making, surveillance, or large-scale data analysis. The societal effects of such downstream applications depend primarily on how future practitioners apply these ideas, including whether appropriate safeguards, fairness considerations, and privacy protections are implemented.

Overall, we believe the primary impact of this work is to advance the mathematical foundations of learning theory and improve understanding of the limits and capabilities of learning under noisy supervision.

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

## A. Deferred preliminaries

**Definition 7** (VC Dimension). *A set of points $S = \{x_1, \ldots, x_d\} \subset \mathcal{X}$ is said to be **shattered** by a concept class $\mathcal{H}$ if for every possible labeling of the points in $S$, there exists a concept $h \in \mathcal{H}$ that realizes this labeling. Formally, for every subset $S' \subseteq S$, there exists an $h \in \mathcal{H}$ such that $h(x) = 1$ for all $x \in S'$ and $h(x) = 0$ for all $x \in S \setminus S'$.*

*The **Vapnik-Chervonenkis (VC) dimension** of a concept class $\mathcal{H}$, denoted $\mathrm{VC}(\mathcal{H})$, is the maximum size of a set of points that can be shattered by $\mathcal{H}$. If arbitrarily large shattered sets exist, the VC dimension is said to be infinite.*

**Definition 8** (Star number). *The **star number** ($\mathfrak{s}$) of a hypothesis class $\mathcal{H}$ is the largest integer $s$ such that there exist distinct points $x_1, \ldots, x_s \in \mathcal{X}$ and classifiers $h_0, \ldots, h_s \in \mathcal{H}$ where for all $i \in [s]$, the disagreement set $DIS(\{h_0, h_i\}) = \{x \in \mathcal{X} : h_0(x) \neq h_i(x)\}$ satisfies $DIS(\{h_0, h_i\}) \cap \{x_1, \ldots, x_s\} = \{x_i\}$. If no such largest integer exists, we define $\mathfrak{s} = \infty$.*

**Definition 9** (Realizable distribution). *A distribution $P$ over $\mathcal{X} \times \{0, 1\}$ is said to be realizable with respect to the (total) concept class $\mathcal{H}$ if $\inf_{h \in \mathcal{H}} er_P(h) = 0$.*

**Definition 10** (Littlestone tree (Bousquet et al., 2021)). *A Littlestone tree for $\mathcal{H}$ of depth $d \leq \infty$ is a collection*

$$\left\{ x_{\mathbf{u}} : 0 \leq k < d, \mathbf{u} \in \{0, 1\}^k \right\} \subseteq \mathcal{X}$$

*such that for every $n < d$ and $\mathbf{y} \in \{0, 1\}^d$, there exists $h \in \mathcal{H}$ so that $h(x_{\mathbf{y} \leq k}) = y_{k+1}$ for $0 \leq k \leq n$, where $\mathbf{y}_{\leq k} = (y_1, ..., y_k)$ for any sequence $\mathbf{y}$.*

**Definition 11** (VCL tree (Bousquet et al., 2021)). *A Vapnik-Chervonenkis-Littlestone (VCL) tree for $\mathcal{H}$ of depth $d \leq \infty$ is a collection:*

$$t = \left\{ x_{\mathbf{u}} \in \mathcal{X}^{k+1} : 0 \leq k < d, \mathbf{u} \in \prod_{j=1}^{k} \{0, 1\}^j \right\}$$

*such that for every $n < d$ and $\mathbf{y} \in \prod_{j=1}^{n+1} \{0, 1\}^j$, there exists a concept $h \in \mathcal{H}$ so that for all $0 \leq i \leq k$ and $0 \leq k \leq n$ we have $h(x_{\mathbf{y}_{\leq k}}^i) = y_{k+1}^i$, where*

$$\mathbf{y}_{\leq k} = \left( y_1^0, \left( y_2^0, y_2^1 \right), \ldots, \left( y_k^0, \ldots, y_k^{k-1} \right) \right) \ and$$
$$x_{\mathbf{y}_{\leq k}} = \left( x_{\mathbf{y}_{\leq k}}^0, \ldots, x_{\mathbf{y}_{\leq k}}^k \right).$$

**Definition 12** (Star tree (Hanneke et al., 2024)). *A star tree for $\mathcal{H}$ of depth $d \leq \infty$ is a collection:*

$$t = \left\{ (x_{\mathbf{u}}, y_{\mathbf{u}}) \in (\mathcal{X} \times \{0, 1\})^{k+2} : 0 \leq k < d, \mathbf{u} \in \prod_{j=1}^{k} \{0, \ldots, j\} \right\}$$

*such that for every $n < d$ and $\mathbf{u} \in \prod_{j=1}^{n+1} \{0, \ldots, j\}$, there exists a concept $h \in \mathcal{H}$ so that for all $0 \leq k \leq n, 0 \leq i \leq k+1$:*

$$h(x_{\mathbf{u}_{\leq k}}^i) = \begin{cases} 1 - y_{\mathbf{u}_{\leq k}}^{u_k} & i = u_k \\ y_{\mathbf{u}_{\leq k}}^i & otherwise \end{cases}$$
$$,where\ u_{\leq k} = (u_1, u_2, ..., u_k).$$

## B. Equivalence of Excess Error under Strong and Weak Massart Conditions

The aim of this section is to justify our choice of excess error notion, namely the excess error with respect to the Bayes optimal classifier $h_P^\star$ (Definition 2), is sufficiently tight for both sample complexity for partial concept classes under the weak Massart noise condition, and for full concept classes under the strong Massart noise condition. We start by showing that under the strong $\beta$-bounded noise condition, the terms $\inf_{h \in \mathcal{H}} er_P(h)$ and $er_P(h_P^\star)$ are equivalent.

**Lemma 7.** *Let $P$ satisfy the strong $\beta$-bounded noise condition with respect to $\mathcal{H}$, then $\inf_{h \in \mathcal{H}} er_P(h) = er_P(h_P^\star)$*

**Proof:** As $\inf_{h \in \mathcal{H}} er_P(h) \geq er_P(h_P^\star)$ holds trivially under the strong $\beta$-bounded noise condition, we only need to show that $\inf_{h \in \mathcal{H}} er_P(h) \leq er_P(h_P^\star)$ also holds.

Let $\{h_t\}_{t \in \mathbb{N}}$ be a sequence of concepts from $\mathcal{H}$ such that $\lim_{t \to \infty} \mathbf{P}[h_t(X) \neq h_P^\star(X)] = 0$. For each $h_t$ we have

$$er(h_t) = \mathbf{P}[h_t(X) \neq Y \wedge h_P^\star(X) = Y] + \mathbf{P}[h_t(X) \neq Y \wedge h_P^\star(X) \neq Y] \leq \mathbf{P}[h_t(X) \neq h_P^\star(X)] + \mathbf{P}[h_P^\star(X) \neq Y]$$

As $\inf_{h \in \mathcal{H}} er_P(h) \leq er_P(h_t)$ for every $t$, taking the limit $t \to \infty$ concludes the proof. $\qquad\square$

We now move on to show the equivalence of the approximation error of the partial concept classes and the Bayes expected error under the weak Massart condition. In contrast to the previous item, this equivalence holds only under certain conditions: when either the marginal distribution $P_\mathcal{X}$ has countable support, or the (partial) concept class $H$ has finite VC dimension. The approximation error is a tighter excess-error measure for partial concept classes (Alon et al., 2021). For a partial concept class $H$, it is defined as $er_P(H) := \lim_{n \to \infty} \varepsilon^\star(n)$, where $\varepsilon^\star(n) := \mathbb{E}_{S \sim P^n}[\min_{h \in H} \hat{er}_S(h)]$.

**Lemma 8.** *Let $P$ satisfy the weak $\beta$-bounded noise condition with respect to the partial concept class $H$, then for every partial concept class $H$ we have $er_P(H) \leq er_P(h_P^\star)$. Conversely, if $P_\mathcal{X}$ has countable support or $\mathrm{VC}(H) \leq d$, then we also have $er_P(H) \geq er_P(h_P^\star)$.*

**Proof:** The first inequality $er_P(H) \leq er_P(h_P^\star)$ is straightforward.

$$er_P(h_P^\star) = \mathbb{E}_{S \sim P^n}[\hat{er}_S(h_P^\star)] \geq \mathbb{E}_{S \sim P^n}\left[\min_{h \in H} \hat{er}_S(h)\right] = er_P(H)$$

Moving to the second part, we first prove the claim for discrete distributions, that is, when $P_\mathcal{X}$ has countable support. To this end, we start with the basic case of $\mathcal{X} = [m]$, where $m \in \mathbb{N}$. Let us assume without loss of generality that $h_P^\star(i) = 1$ for every $i \in [m]$. and define $N_S(x, y)$ to be the number of samples of $(x, y)$ given a sample $S \subset (X, Y)^*$, where we also write $N_S(x) := N_S(x, 0) + N_S(x, 1)$. Let $\{A_n\}_{n \geq 1}$ be a sequence of events, where

$$A_n = \{\exists i \in [m] \text{ such that } N_S(i, 0) > N_S(i, 1), \text{ where } S \sim P^n\}$$

First, we start by claiming that $\lim_{n \to \infty} \mathbf{P}[A_n] = 0$. Indeed, we start by applying union bound we have that $\mathbf{P}[A_n] \leq \sum_{i \in [m]} \mathbf{P}[N_S(i, 0) > N_S(i, 1)]$. Fix some $i \in [m]$, by conditioning on the event that $N_S(i) = k$ we may use Hoeffding inequality to obtain the following bound.

$$\mathbf{P}[N_S(i, 0) > N_S(i, 1) | N_S(i) = k] \leq \exp\left(-2k\left(\frac{1}{2} - \beta\right)^2\right) = C^k$$

where we set $C := \exp\left(-2(\frac{1}{2} - \beta)^2\right)$. Thus, using law of total probability, we obtain the probability generating function $G(C) := \mathbb{E}[C^Z]$, where $Z$ is a Binomial random variable with $n$ trials, each with probability $p_i = \mathbf{P}[X = i]$, which is known to be $[(1 - p_i) + p_i C]^n$. Therefore, we can conclude that

$$\mathbf{P}[A_n] \leq \sum_{i \in [m]} [(1 - p_i) + p_i C]^n$$

As we have $C < 1$, by taking the limit, the claim follows.

To finalize the argument for discrete distributions, let us denote $h_S$ as the minimizer of a given sample $S$, that is $h_S := \mathrm{argmin}_{h \in H} \hat{er}_S(h)$. Observe that for every $S \sim P^n$ we have

$$\mathbb{E}[\hat{er}_S(h_S)] \geq \mathbb{E}[\hat{er}_S(h_S) \mathbb{1}_{\{\bar{A}_n\}}] = \mathbb{E}[\hat{er}_S(h_P^\star) \mathbb{1}_{\{\bar{A}_n\}}]$$

By applying the dominated convergence theorem on the sequence of random variables $\left\{\hat{er}_S(h_P^\star) \mathbb{1}_{\{\bar{A}_n\}}\right\}_{n \geq 1}$, we have that $\mathbb{E}[\hat{er}_S(h_P^\star) \mathbb{1}_{\{\bar{A}_n\}}] = \mathbb{E}[\hat{er}_S(h_P^\star)]$.

To extend this argument to countable support, we enumerate the support as $\mathcal{X} = \{x_1, x_2, \ldots\}$, and define the sequence of events $\{B_m\}_{m \geq 1}$, where the event $B_m$ is defined as the support of the sample $S$ being a subset of $\{x_1, \ldots, x_m\}$. Using the previous claim for finite support, we have

$$\mathbb{E}[\hat{er}_S(h_S)] \geq \mathbb{E}[\hat{er}_S(h_S) \mathbb{1}_{\{B_m\}}] \geq \mathbb{E}[\hat{er}_S(h_P^\star) \mathbb{1}_{\{B_m\}}]$$

using the dominated convergence theorem again on the sequence of random variables $\left\{\hat{er}_S(h_P^\star)\mathbb{1}_{\{B_m\}}\right\}_{m \geq 1}$, the first part of our proof is complete.

Next, we move to the case of $\mathrm{VC}(H) = d < \infty$. Our proof is a simple extension to (Alon et al., 2021, Lemma 39), which establishes the equivalence of $er_P(\mathcal{H})$ and $\inf_{h \in \mathcal{H}} er_P(h)$ for total concept classes with finite VC dimension. Let $H[X_1, \ldots, X_n]$ be the projection of $H$ onto the random variables $X_1, \ldots, X_n$, and let $\tilde{H} := \tilde{H}[X_1, \ldots, X_n]$ be the disambiguation of $H[X_1, \ldots, X_n]$ (that is, the completion of every partial concept in $H[X_1, \ldots, X_n]$ such that it is defined on $X_1, \ldots, X_n$). By (Alon et al., 2021, Theorem 12), there must exist such a disambiguation of size at most $O(n^{d \log(n)})$. Thus, by conditioning on $\mathcal{X}_S$, the support of $X_1, \ldots, X_n$, we apply uniform convergence and get that with probability at least $1 - \delta$, for every $\tilde{h} \in \tilde{H}$, the following holds:

$$\min_{h \in \tilde{H}} \hat{er}_S(h) \geq \min_{h \in \tilde{H}} er_{P|\mathcal{X}_S}(h) + \sqrt{\frac{d \log(n) + \log(1/\delta)}{n}}$$

Observe that for any $n$ the minimizer of $er_{P|\mathcal{X}_S}(h)$ over $h \in \tilde{H}$ is $h_P^\star$. By setting $\delta = O(n^{-d})$ and taking expectation, by the law of total expectation we

$$\mathbb{E}\left[\min_{h \in H} \hat{er}_S(h)\right] \geq er_P(h_P^\star) - O\left(\sqrt{\frac{d \log(n)}{n}}\right)$$

Letting $n \to \infty$ concludes the proof. $\qquad\square$

## C. Excess Error Bound of the Transductive Learning Algorithm

The section is dedicated to the proof of the expected error bound of the TERM algorithm for $\beta$-bounded distributions (Theorem 1). For the rest of this section, let $H$ be a partial concept class over a non-empty set $\mathcal{X}$, and let $P$ be a distribution over $\mathcal{X} \times \{0, 1\}$ such that it satisfies Definition 4 with respect to $H$. We denote the set $H[x_1, ..., x_n]$ as the *realizable* version space of $H$ over $(x_1, \ldots, x_n)$, that is, $H[x_1, ..., x_n] = \left\{f \in \{0, 1\}^{(x_1, \ldots, x_n)} : \exists h \in H \text{ such that } f(x_i) = h(x_i) \text{ for all } i \in [n]\right\}$. We will also use $\hat{H}[x_1, ..., x_n]$ to denote the version space of $H$ without dropping the partial concepts for the projection. Before we go to the proof of Theorem 1, we first review the auxiliary lemmas used throughout the proof. The first lemma we will use is the simplified variant contraction lemma of (Zhivotovskiy & Hanneke, 2018), allowing us to strip the labels so that we can proceed to analyze the empirical functional structure of the concept class.

**Lemma 9** (Contraction). *(Zhivotovskiy & Hanneke, 2018) Let $\{(X_1, Y_1), ..., (X_n, Y_n)\} \sim P^n$ and $\varepsilon_1, ..., \varepsilon_n \sim \text{Uniform}\left(\{-1, 1\}\right)^n$ be independent from each other. For any $c \in [0, 1]$, there exists random variables $\xi_1, ..., \xi_n$ conditionally independent given $X_1, ..., X_n$ with $|\xi_i| \leq 1$, $\mathbb{E}[\xi_i | X_1, ..., X_n] = 0$ and $\mathbb{E}[e^{\lambda \xi_i} | X_1, ..., X_n] \leq e^{\lambda^2/2}$ for all $\lambda \in \mathbb{R}$, such that*

$$\mathbb{E}\left[\max_{h \in H[X_1, ..., X_n]} \sum_{i=1}^n (\varepsilon_i - c)\left(\mathbb{1}_{\{h(X_i) \neq Y_i\}} - \mathbb{1}_{\{h_P^\star(X_i) \neq Y_i\}}\right)\right]$$

$$\leq 3\mathbb{E}\left[\max_{h \in H[X_1, ..., X_n]} \sum_{i=1}^n \left(\xi_i - (1 - 2\beta)\frac{c}{3}\right)\mathbb{1}_{\{h(X_i) \neq h_P^\star(X_i)\}}\right]$$

We emphasize that the original statement of Lemma 9 was stated for a class of total functions. However, the result there holds even under the conditional expectation given $X_1, ..., X_n$, and thus it can be applied to the *total* concept class $H[X_1, ..., X_n]$. We emphasize that the statement above is also slightly different (simpler) than the original result, and therefore we include a proof of this lemma in Subsection C.1 for completeness.

Next, we import some definitions from (Zhivotovskiy & Hanneke, 2018). For any $n \in \mathbb{N}$ and sequence $x_1, ..., x_n \in \mathcal{X}$, and any $G \subset H[x_1, ..., x_n]$ and $\varepsilon \geq 0$, denote by $\mathcal{M}_1(G, \varepsilon, \{x_1, ..., x_n\})$ the maximum size of a subset $G_\varepsilon \subseteq G$ such that every distinct $g, g' \in G_\varepsilon$ have $\sum_{i=1}^n \mathbb{1}_{\{g(x_i) \neq g'(x_i)\}} > \varepsilon$: that is, a maximum $\varepsilon$-packing under the Hamming distance. For any function $h : \{x_1, ..., x_n\} \to \{0, 1\}$, and value $\gamma \geq 0$, we denote the $\gamma$-radius ball under Hamming distance by $B(h, \gamma, \{x_1, ..., x_n\})$. That is,

$$B(h, \gamma, \{x_1, ..., x_n\}) = \left\{h' \in H[x_1, ..., x_n] : \sum_{i=1}^n \mathbb{1}_{\{h'(x_i) \neq h(x_i)\}} \leq \gamma\right\}.$$

We are now ready for the definition of local metric entropy.

**Definition 13** (Local metric entropy). *For any $n \in \mathbb{N}, \gamma \geq 0$ and $c > 0$, define the local empirical packing number:*

$$\mathcal{M}_1^{loc}(H, \gamma, n, c) = \max_{x_1, ..., x_n} \max_{h \in H[x_1, ..., x_n]} \max_{\varepsilon \geq \gamma} \mathcal{M}_1(B(h, \varepsilon/c, \{x_1, ..., x_n\}), \varepsilon/2, \{x_1, ..., x_n\})$$

*and for any $c, c' > 0$, define the local metric entropy:*

$$\gamma_{c,c'}^{loc}(n, H) = \max \left\{ \gamma \in \mathbb{N} : c\gamma \leq \log \left( \mathcal{M}_1^{loc}(H, \gamma, n, c') \right) \right\}$$

We emphasize that both the $\gamma$-radius ball and local empirical packing number are taken with respect to **realizable projections only** and are well-defined as these projections are almost surely non-empty. Moving on, this next localization lemma bounds the offset sub-gaussian process with the local metric entropy.

**Lemma 10** (Localization). *(Zhivotovskiy & Hanneke, 2018) Let $c \in [0, 1/4]$ be a constant, and fix any $x_1, ..., x_n \in \mathcal{X}$ such that $\exists h \in H[x_1, ..., x_n]$ with $\sum_{i=1}^n \mathbb{1}_{\{h(x_i) \neq h_P^\star(x_i)\}} = 0$. Let $\xi_1, ..., \xi_n$ be independent random variables with $|\xi_i| \leq 1$ and with $\mathbb{E}[\xi_i] = 0$ and $\mathbb{E}[e^{\lambda \xi_i}] \leq e^{\lambda^2/2}$ for all $\lambda \in \mathbb{R}$. Then*

$$\mathbb{E}\left[ \max_{h \in H[x_1, ..., x_n]} \frac{1}{n} \sum_{i=1}^n (\xi_i - 4c) \mathbb{1}_{\{h(x_i) \neq h_P^\star(x_i)\}} \right] = O\left( \frac{\gamma_{c,c}^{loc}(n, H)}{n} \right).$$

The last result we bring shows the relation of the local metric entropy with the VC dimension and star number, respectively.

**Lemma 11** (Bound on local metric entropy). *(Zhivotovskiy & Hanneke, 2018) For any constant $c > 0$ and any $n \in \mathbb{N}$ with $\sqrt{\frac{d}{n}} < c$,*

$$\gamma_{c,c}^{loc}(n, H) = O\left( \frac{d \log \left( \min \left\{ \frac{nc^2}{d}, \mathfrak{s} \right\} \right)}{c} + \frac{d \log \left( \frac{1}{c} \right)}{c} \right).$$

While the original statement of this result by (Zhivotovskiy & Hanneke, 2018) was for total concept class, it is straightforward to note that it necessarily also holds for partial concept classes, since for any $\gamma \in \mathbb{N}$ satisfying $c\gamma \leq \log \left( \mathcal{M}_1^{loc}(H, \gamma, n, c) \right)$, for the sequence $x_1, ..., x_n$ realizing the maximum value in the definition of $\mathcal{M}_1^{loc}(H, \gamma, n, c)$, we have $\mathcal{M}_1^{loc}(H, \gamma, n, c) = \mathcal{M}_1^{loc}(H[x_1, ..., x_n], \gamma, n, c)$, so that the right hand side in the lemma statement is an upper bound on this $\gamma$ (noting that the VC dimension and star number of $H[x_1, ..., x_n]$ is at most the VC dimension and star number, respectively, of the partial concept class $H$)[7], and this being true for all such $\gamma$, it is an upper bound on $\gamma_{c,c}^{loc}(n, H)$. We are now ready for the proof of Theorem 1.

**Proof:**

We start by applying the exchangeability argument to obtain the expected empirical excess error, matching the size of the sample used by the empirical risk minimizer.

$$\mathbb{E}\left[ er_P(\hat{h}_n) \right] - er_P(h_P^\star) = \mathbb{E}\left[ \mathbb{1}_{\{\hat{h}_n(X_{n+1}) \neq Y_{n+1}\}} - \mathbb{1}_{\{h_P^\star(X_{n+1}) \neq Y_{n+1}\}} \right]$$

$$= \mathbb{E}\left[ \frac{1}{\lceil n/2 \rceil} \sum_{i \in [\lceil n/2 \rceil]} \left( \mathbb{1}_{\{\hat{h}_n(X_{\lceil n/2 \rceil + i}) \neq Y_{\lceil n/2 \rceil + i}\}} - \mathbb{1}_{\{h_P^\star(X_{\lceil n/2 \rceil + i}) \neq Y_{\lceil n/2 \rceil + i}\}} \right) \right]$$

Denote $S = \{(X_i, Y_i)\}_{i \in [\lceil n/2 \rceil]}$ and $S' = \{(X_{\lceil n/2 \rceil + i}, Y_{\lceil n/2 \rceil + i})\}_{i \in [\lceil n/2 \rceil]}$, then for any classifier $h$, we define the following

$$\hat{er}_S(h) = \frac{1}{\lceil n/2 \rceil} \sum_{i \in [\lceil n/2 \rceil]} \mathbb{1}_{\{h(X_i) \neq Y_i\}}, \qquad \hat{er}_{S'}(h) = \frac{1}{\lceil n/2 \rceil} \sum_{i \in [\lceil n/2 \rceil]} \mathbb{1}_{\{h(X_{\lceil n/2 \rceil + i}) \neq Y_{\lceil n/2 \rceil + i}\}}$$

---

[7] This holds because for every $x_1, ..., x_n$ with $n \in \mathbb{N}$ we have $\text{VC}(H[x_1, ..., x_n]) \leq \text{VC}(\hat{H}[x_1, ..., x_n]) \leq \text{VC}(H)$. Specifically, restricting the domain of a concept class or limiting it to a realizable subclass can only decrease the size of the maximum shattered set, meaning the VC dimension is upper-bounded by that of the original class. An identical rationale holds for the star number.

Then the last line above equals $\mathbb{E}[\hat{er}_{S'}(\hat{h}_n) - \hat{er}_{S'}(h_P^\star)]$. Thus, it suffices to show that the expectation of the empirical excess error is bounded from above by $O\left(\frac{d\log(\min\{n/d,\mathsf{s}\})}{n}\right)$. We will apply the *offset Rademacher complexity* technique from Zhivotovskiy and Hanneke (2018). Specifically, let $c \geq 0$ be any constant. For brevity, we make the following notations. Denote $\hat{H} = H[X_1, ..., X_{n+1}]$, and for any $h \in \hat{H}$, we define the pointwise loss difference

$$\ell_i(h) := \mathbb{1}_{\{h(X_i) \neq Y_i\}} - \mathbb{1}_{\{h_P^\star(X_i) \neq Y_i\}}$$

and the corresponding empirical excess error

$$\hat{\mathcal{E}}_S(h) := \hat{er}_S(h) - \hat{er}_S(h_P^\star), \qquad \hat{\mathcal{E}}_{S'}(h) := \hat{er}_{S'}(h) - \hat{er}_{S'}(h_P^\star).$$

The two latter are the excess risk on first and second halves of the random sample, respectively. Since $\hat{h}_n$ is the ERM on $\{(X_i, Y_i)\}_{i \in [[n/2]]}$, we have $\hat{\mathcal{E}}_S(\hat{h}_n) \leq 0$ almost surely. Thus,

$$\mathbb{E}\left[\hat{er}_{S'}(\hat{h}_n) - \hat{er}_{S'}(h_P^\star)\right] = \mathbb{E}\left[\hat{\mathcal{E}}_{S'}(\hat{h}_n)\right] \leq$$

$$\mathbb{E}\left[\hat{\mathcal{E}}_{S'}(\hat{h}_n) - (1+c)\hat{\mathcal{E}}_S(\hat{h}_n)\right] \leq \mathbb{E}\left[\max_{h \in \hat{H}} \hat{\mathcal{E}}_{S'}(h) - (1+c)\hat{\mathcal{E}}_S(h)\right] =$$

$$\mathbb{E}\left[\max_{h \in \hat{H}}\left(1 + \frac{c}{2}\right)\left(\hat{\mathcal{E}}_{S'}(h) - \hat{\mathcal{E}}_S(h)\right) - \frac{c}{2}\left(\hat{\mathcal{E}}_{S'}(h) + \hat{\mathcal{E}}_S(h)\right)\right]$$

Recalling the definitions of $\hat{\mathcal{E}}_S(h)$ and $\hat{\mathcal{E}}_{S'}(h)$, this last expression equals

$$\mathbb{E}\left[\max_{h \in \hat{H}}\left(1 + \frac{c}{2}\right)\left(\frac{1}{\lceil n/2 \rceil}\sum_{i \in [[n/2]]}\left(\ell_i(h) - \ell_{\lceil n/2 \rceil + i}(h)\right)\right) - \frac{c}{2\lceil n/2 \rceil}\left(\sum_{i \in [2\lceil n/2 \rceil]}\ell_i(h)\right)\right] \tag{1}$$

Now, consider the conditional distribution of the random variables in the expectation above, conditioned on the random variables $\{\{X_i, X_{\lceil n/2 \rceil + i}\}\}_{i=1}^{\lceil n/2 \rceil}$, which denotes the sequence of *unordered* sets $\{X_i, X_{\lceil n/2 \rceil + i}\}$. In particular, conditioned on this sequence $\{\{X_i, X_{\lceil n/2 \rceil + i}\}\}_{i=1}^{\lceil n/2 \rceil}$, the $i$-th index of the sample is equally likely to be $X_i$ or $X_{\lceil n/2 \rceil + i}$. Therefore, for any $\varepsilon_1, ..., \varepsilon_{\lceil n/2 \rceil} \in \{-1, 1\}$, the random variables

$$\max_{h \in \hat{H}}\left(1 + \frac{c}{2}\right)\left(\frac{1}{\lceil n/2 \rceil}\sum_{i \in [[n/2]]}\left(\ell_i(h) - \ell_{\lceil n/2 \rceil + i}(h)\right)\right) - \frac{c}{2\lceil n/2 \rceil}\left(\sum_{i \in [2\lceil n/2 \rceil]}\ell_i(h)\right)$$

and

$$\max_{h \in \hat{H}}\left(1 + \frac{c}{2}\right)\left(\frac{1}{\lceil n/2 \rceil}\sum_{i \in [[n/2]]}\varepsilon_i\left(\ell_i(h) - \ell_{\lceil n/2 \rceil + i}(h)\right)\right) - \frac{c}{2\lceil n/2 \rceil}\left(\sum_{i \in [2\lceil n/2 \rceil]}\ell_i(h)\right)$$

have the same conditional distribution given $\{\{X_i, X_{\lceil n/2 \rceil + i}\}\}_{i=1}^{\lceil n/2 \rceil}$. Letting $\varepsilon_1, ..., \varepsilon_{\lceil n/2 \rceil}$ be independent Rademacher

random variables (that is, $\mathbf{P}\left(\varepsilon_i = 1\right) = \mathbf{P}\left(\varepsilon_i = -1\right) = 1/2$) independent of $S$, we have that Equation (1) is equal

$$\mathbb{E}\left[\max_{h\in\hat{H}}\left(1+\frac{c}{2}\right)\left(\frac{1}{\lceil n/2\rceil}\sum_{i\in[[n/2]]}\varepsilon_i\left(\ell_i(h)-\ell_{\lceil n/2\rceil+i}(h)\right)\right)-\frac{c}{2\lceil n/2\rceil}\left(\sum_{i\in[2\lceil n/2\rceil]}\ell_i(h)\right)\right]$$

$$=\left(1+\frac{c}{2}\right)\mathbb{E}\left[\max_{h\in\hat{H}}\left(\frac{1}{\lceil n/2\rceil}\sum_{i\in[[n/2]]}\varepsilon_i\left(\ell_i(h)-\ell_{\lceil n/2\rceil+i}(h)\right)\right)-\frac{c}{(2+c)\lceil n/2\rceil}\left(\sum_{i\in[2\lceil n/2\rceil]}\ell_i(h)\right)\right]$$

$$=\left(1+\frac{c}{2}\right)\mathbb{E}\left[\max_{h\in\hat{H}}\left(\frac{1}{\lceil n/2\rceil}\sum_{i\in[[n/2]]}\left(\varepsilon_i-\frac{c}{2+c}\right)\ell_i(h)\right)\right.$$

$$\left.+\left(\frac{1}{\lceil n/2\rceil}\sum_{i\in[[n/2]]}\left((-\varepsilon_i)-\frac{c}{2+c}\right)\ell_{\lceil n/2\rceil+i}(h)\right)\right]$$

$$\leq\left(1+\frac{c}{2}\right)\mathbb{E}\left[\max_{h\in\hat{H}}\left(\frac{1}{\lceil n/2\rceil}\sum_{i\in[[n/2]]}\left(\varepsilon_i-\frac{c}{2+c}\right)\ell_i(h)\right)\right]+$$

$$\left(1+\frac{c}{2}\right)\mathbb{E}\left[\max_{h\in\hat{H}}\left(\frac{1}{\lceil n/2\rceil}\sum_{i\in[[n/2]]}\left((-\varepsilon_i)-\frac{c}{2+c}\right)\ell_{\lceil n/2\rceil+i}(h)\right)\right]$$

and since $(-\varepsilon_1,..,-\varepsilon_{n/2})$ has the same distribution as $(\varepsilon_1,...,\varepsilon_{n/2})$ (and both are independent of $S$), and the exchangeability of the sequence $(X_1,Y_1),\ldots,(X_{2\lceil n/2\rceil},Y_{2\lceil n/2\rceil})$ implies that $((X_1,Y_1),...,(X_{\lceil n/2\rceil},Y_{\lceil n/2\rceil}))$ and $((X_{\lceil n/2+1\rceil},Y_{\lceil n/2+1\rceil}),\ldots,(X_{2\lceil n/2\rceil},Y_{2\lceil n/2\rceil}))$ both have the same distribution as $((X_1,Y_1),...,(X_{n/2},Y_{n/2}))$, this last expression above is equal

$$2\left(1+\frac{c}{2}\right)\mathbb{E}\left[\frac{1}{\lceil n/2\rceil}\max_{h\in\hat{H}}\sum_{i\in[[n/2]]}\left(\varepsilon_i-\frac{c}{2+c}\right)\ell_i(h)\right]$$

$$=2\left(\frac{2+c}{\lceil n/2\rceil}\right)\mathbb{E}\left[\max_{h\in\hat{H}}\sum_{i\in[[n/2]]}\left(\varepsilon_i-\frac{c}{2+c}\right)\ell_i(h)\right]$$

$$\leq2\left(\frac{2+c}{\lceil n/2\rceil}\right)\mathbb{E}\left[\max_{h\in H[X_1,...,X_{\lceil n/2\rceil}]}\sum_{i\in[[n/2]]}\left(\varepsilon_i-\frac{c}{2+c}\right)\ell_i(h)\right]$$

where the last inequality is due to the fact that $\hat{H}[X_1,...,X_{\lceil n/2\rceil}] \subseteq H[X_1,...,X_{\lceil n/2\rceil}]$: that is, the patterns on $X_1,...,X_{\lceil n/2\rceil}$ realizable by $H$ can only be a superset of the patterns on $X_1,...,X_{\lceil n/2\rceil}$ present among the patterns on $X_1,...,X_{2\lceil n/2\rceil}$ realizable by $H$ (indeed, it may sometimes be a strict superset, since $\hat{H}[X_1,...,X_{\lceil n/2\rceil}]$ only contains the patterns on $X_1,..,X_{\lceil n/2\rceil}$ realizable by partial concepts in $H$ whose support also includes $X_{\lceil n/2\rceil+1},...,X_{2\lceil n/2\rceil}$, an additional restriction not required for the patterns in $H[X_1,...,X_{\lceil n/2\rceil}]$). By Lemma 9, the last line above is at most

$$6\left(\frac{2+c}{\lceil n/2\rceil}\right)\mathbb{E}\left[\max_{h\in H[X_1,...,X_{\lceil n/2\rceil}]}\sum_{i\in[[n/2]]}\left(\xi_i-(1-2\beta)\frac{c}{3(2+c)}\right)\mathbb{1}_{\left\{h(X_i)\neq h_P^*(X_i)\right\}}\right]$$

where $\xi_1,...,\xi_{\lceil n/2\rceil}$ are conditionally independent given $X_1,...,X_{\lceil n/2\rceil}$ where for each $i \in [[n/2]]$ we have $|\xi_i| \leq 1$, $\mathbb{E}\left[\xi_i|X_1,...,X_{\lceil n/2\rceil}\right]$, and $\mathbb{E}\left[e^{\lambda\xi_i}|X_1,...,X_{n/2}\right] < e^{\lambda^2/2}$ for any $\lambda \in \mathbb{R}$.

Applying Lemma 10 under the conditional distribution given $X_1,...,X_{\lceil n/2\rceil}$, together with the law of total expectation,

implies

$$6\left(\frac{2+c}{\lceil n/2\rceil}\right)\mathbb{E}\left[\max_{h\in H[X_1,\ldots,X_{\lceil n/2\rceil}]}\sum_{i\in[[n/2]]}\left(\xi_i-\frac{c(1-2\beta)}{3(2+c)}\right)\mathbb{1}_{\left\{h(X_i)\neq h_P^\star(X_i)\right\}}\right]=O\left(\frac{\gamma_{c',c'}^{loc}(n/2,H)}{n}\right).$$

where $c'=(1-2\beta)\frac{c}{12(2+c)}$. In particular, applying this result with $c=1$, together with Lemma 11, yields the required bound of $O\left(\frac{d\log\left(\min\left\{\frac{n(1-2\beta)^2}{d},\mathfrak{s}\right\}\right)}{(1-2\beta)n}+\frac{d\log\left(\frac{1}{1-2\beta}\right)}{(1-2\beta)n}\right)=O\left(\frac{d\log\left(\min\left\{\frac{n(1-2\beta)}{d},\frac{\mathfrak{s}}{1-2\beta}\right\}\right)}{(1-2\beta)n}\right).$

$\square$

## C.1. Proof of Lemma 9

Denote by $\beta_i=\mathbf{P}\left[h_P^\star(X_i)\neq Y_i|X_i\right]$ for $i\in[n]$, and note that $\beta_i\leq\beta$ almost surely (where $\beta<1/2$ is as in Definition 4, as satisfied by the distribution $P$). From the definition of $h_P^\star$ we have

$$\mathbb{E}\left[\max_{h\in H[X_1,\ldots,X_n]}\sum_{i=1}^n(\varepsilon_i-c)\left(\mathbb{1}_{\{h(X_i)\neq Y_i\}}-\mathbb{1}_{\left\{h_P^\star(X_i)\neq Y_i\right\}}\right)\right]$$

$$=\mathbb{E}\left[\max_{h\in H[X_1,\ldots,X_n]}\sum_{i=1}^n(\varepsilon_i-c)(1-2\mathbb{1}_{\left\{h_P^\star(X_i)\neq Y_i\right\}})\mathbb{1}_{\left\{h(X_i)\neq h_P^\star(X_i)\right\}}\right]$$

Now observe that for all $i\in[n]$ we have $1-2\mathbb{1}_{\left\{h_P^\star(X_i)\neq Y_i\right\}}\in\{-1,1\}$, therefore the random variables $\{\varepsilon_i(1-2\mathbb{1}_{\left\{h_P^\star(X_i)\neq Y_i\right\}})\}_{i=1}^n$ remains distributed as Uniform$(\{-1,1\})^n$ and independent of $\{(X_i,Y_i)\}_{i=1}^n$ just as $\{\varepsilon_i\}_{i=1}^n$ is. Thus, the above expression can be rewritten as

$$\mathbb{E}\left[\max_{h\in H[X_1,\ldots,X_n]}\sum_{i=1}^n(\varepsilon_i-c(1-2\mathbb{1}_{\left\{h_P^\star(X_i)\neq Y_i\right\}}))\mathbb{1}_{\left\{h(X_i)\neq h_P^\star(X_i)\right\}}\right]$$

$$=\mathbb{E}\left[\max_{h\in H[X_1,\ldots,X_n]}\sum_{i=1}^n(\varepsilon_i+c((1-2\beta_i)-(1-2\mathbb{1}_{\left\{h_P^\star(X_i)\neq Y_i\right\}}))-(1-2\beta_i)c)\mathbb{1}_{\left\{h(X_i)\neq h_P^\star(X_i)\right\}}\right]$$

$$\leq\mathbb{E}\left[\max_{h\in H[X_1,\ldots,X_n]}\sum_{i=1}^n(\varepsilon_i+c((1-2\beta_i)-(1-2\mathbb{1}_{\left\{h_P^\star(X_i)\neq Y_i\right\}}))-(1-2\beta)c)\mathbb{1}_{\left\{h(X_i)\neq h_P^\star(X_i)\right\}}\right].$$

Now let $\xi_i=(1/3)(\varepsilon_i+c((1-2\beta_i)-(1-2\mathbb{1}_{\left\{h_P^\star(X_i)\neq Y_i\right\}})))$. Note that $\xi_i$ are conditionally independent given $X_1,\ldots,X_n$, with $\mathbb{E}[\xi_i|X_1,\ldots,X_n]=0$, and since $-1\leq(1/3)(-1-c+(1-2\beta_i)c)\leq\xi_i\leq(1/3)(1+c+(1-2\beta_i)c)\leq1$, Hoeffding's lemma implies any $\lambda\in\mathbb{R}$ has $\mathbb{E}[e^{\lambda\xi_i}|X_1,\ldots,X_n]\leq e^{\lambda^2/2}$. The claimed inequality immediately follows.

# D. Proof of Theorem 2

Our proof is similar in spirit to that of Alon et al. (2021, Theorem 34), which proves the sample complexity of partial concept classes in the realizable case. Starting with the lower bound, we aim to use the existing lower bound for total concept classes for $\beta$-bounded distributions of (Hanneke, 2016, Theorem 17). Recalling that $Log(x)=\ln(x\vee e)$, this bound can be written in the following form

$$er(\hat{h})-\inf_{h\in\mathcal{H}}er(h)\gtrsim\frac{(d+Log(1/\delta))\vee\beta Log(\mathfrak{s}\wedge(1-2\beta)^2n)}{(1-2\beta)n}\wedge(1-2\beta) \tag{2}$$

More specifically, let $k \leq \mathrm{VC}(H)$, we choose a set of $k$ points in $\mathcal{X}$, $\mathcal{X}_k = \{x_1, \ldots, x_k\}$ that is shattered by $H$, and denote $\mathcal{H}_k$ to be the set of all total functions over $\mathcal{X}_k$. Note that any distribution $P$ over $\mathcal{X}_k \times \{0,1\}$ that is $\beta$-bounded with respect to $\mathcal{H}_k$ (see Definition 1), can be extended to $\mathcal{X} \times \{0,1\}$ by letting $P\left[(\mathcal{X} \setminus \mathcal{X}_k) \times \{0,1\}\right] = 0$, such that it is $\beta$-bounded with respect to $H$ (see definition 4). Thus, any lower bound on sample complexity of PAC learning for the total concept class $\mathcal{H}_k$ will also be a lower bound for PAC learning the partial concept class $H$. Therefore, the bound in 2 implies a sample complexity lower bound of $\Omega\left(\frac{k + \log(1/\delta)}{(1-2\beta)\varepsilon}\right)$. As this implication for $H$ holds for every $k \leq \mathrm{VC}(H)$, we have that $\Omega\left(\frac{d + \log(1/\delta)}{(1-2\beta)\varepsilon}\right)$ for $\mathrm{VC}(H) \leq d$. Otherwise, the partial concept class $H$ is not learnable at all.

Next, we move on to the complementary lower bound. Let $k' \leq \mathfrak{s}$, this time we select a subset of $\mathcal{X}$, $\mathcal{X}_{k'} = \{x_1, \ldots, x_{k'}\}$, which is a star set of $H$, and denote $\mathcal{H}_{k'}$ to be the set of all total concepts from the restriction of $H$ to $\mathcal{X}_{k'}$. By the same reasoning we may apply the complementary excess risk lower bound, which implies sample complexity of $\Omega\left(\frac{\beta \log\left(\min\{k', 1/\varepsilon\}\right)}{\varepsilon(1-2\beta)}\right)$.[8] As this argument holds for every $k' \leq \mathfrak{s}$, we may conclude that $H$ requires sample complexity of $\Omega\left(\frac{\beta \log(\min\{\mathfrak{s}, 1/\varepsilon\})}{\varepsilon(1-2\beta)}\right)$. Taking the maximum of the two, we conclude that $\mathcal{M}(\varepsilon, \delta) = \Omega\left(\frac{d + \beta \log(\min\{\mathfrak{s}, 1/\varepsilon\}) + \log(1/\delta)}{\varepsilon(1-2\beta)}\right)$.

Finally, we proceed to prove the upper bound. We retrace the same steps from Haussler et al. (1994); Long (2001); Alon et al. (2021), converting an algorithm with bounded error rate into an algorithm with PAC guarantees by using confidence boosting. We do so by first splitting one part of the sample into sufficiently large batches, so that at least one of the minimizers of those batches will be $\varepsilon$-accurate classifier with high probability, then use the second part to filter out this classifier using Chernoff and Bernstein inequalities together with the union bound. Given Theorem 1, let us first select the required batch size such that $\mathbb{E}[er_P(\hat{h}_b)] - er_P(h_P^\star) \leq \varepsilon/4$. Specifically, we select our batch size to be $b = \left\lceil C \min\left(\frac{4d \log\left(\frac{\mathfrak{s}}{1-2\beta}\right)}{(1-2\beta)\varepsilon}, \frac{d \log\left(\frac{2}{\varepsilon}\right)}{(1-2\beta)\varepsilon}\right)\right\rceil$, where $C > 0$ is the universal constant from the bound of Theorem 1.

We then select our required sample size to be $n = b\left\lceil \log_2\left(\frac{2}{\delta}\right)\right\rceil + \left\lceil \frac{32}{\varepsilon(1-2\beta)} \log\left(\frac{2\lceil \log_2(2/\delta)\rceil}{\delta}\right)\right\rceil$. For every $i \in \left[\left\lceil \log_2\left(\frac{2}{\delta}\right)\right\rceil\right]$, we set $S_i := \left(X_{1+b(i-1)}, Y_{1+b(i-1)}, \ldots, X_{bi}, Y_{bi}\right)$, and define $h_i(\cdot) = \mathbb{A}(S_i, \cdot)$, where $\mathbb{A}$ is the algorithm from Theorem 1. Let $\tilde{S} = S \setminus \left\{\cup_{i \in \lceil \log_2\left(\frac{2}{\delta}\right)\rceil} S_i\right\}$, we select our output to be $h_{i^*}(X)$, where $i^* := \mathrm{argmin}_{i \in \lceil \log_2\left(\frac{2}{\delta}\right)\rceil} \hat{er}_{\tilde{S}}(h_i)$.

As $\mathbb{E}[er(h_i)] - er(h_P^\star) \leq \frac{\varepsilon}{4}$, we have by Markov that $\mathbf{P}\left[er(h_i) - er(h_P^\star) > \frac{\varepsilon}{2}\right] \leq \frac{1}{2}$. Let us denote by $B_1$ the event that all the classifiers will have excess risk greater than $\frac{\varepsilon}{2}$, then the probability that $B_1$ occur is at most $\frac{1}{2}^{\lceil \log_2\left(\frac{2}{\delta}\right)\rceil} \leq \frac{\delta}{2}$. Let $A$ be the set of all classifiers, that is, $\left\{h_i, i \in \left[\left\lceil \log_2\left(\frac{2}{\delta}\right)\right\rceil\right]\right\}$, let us define the sets $A_{good} := \left\{h \in A : er(h) - er(h_P^\star) \leq \frac{\varepsilon}{2}\right\}$, while $A_{bad} := \{h \in A : er(h) - er(h_P^\star) > \varepsilon\}$.

Fix some $h \in A_{good}$. Observe that

$$\mathbf{P}\left[\hat{er}_{\tilde{S}}(h) - \hat{er}_{\tilde{S}}(h_P^\star) > \frac{3\varepsilon}{4}\right] \leq \mathbf{P}\left[\hat{er}_{\tilde{S}}(h) - \hat{er}_{\tilde{S}}(h_P^\star) - (er(h) - er(h_P^\star)) > \frac{\varepsilon}{4}\right].$$

Define $Z_i = \mathbb{1}_{\{h(X_i) \neq Y_i\}} - \mathbb{1}_{\{h_P^\star(X_i) \neq Y_i\}}$, then $Z_i$ is a random variable such that $|Z_i| \leq 1$ almost surely.

Now, let $m := |\tilde{S}| = \left\lceil \frac{32}{\varepsilon(1-2\beta)} \log\left(\frac{2\lceil \log_2(2/\delta)\rceil}{\delta}\right)\right\rceil$ then $\hat{er}_{\tilde{S}}(h) - \hat{er}_{\tilde{S}}(h_P^\star) - (er(h) - er(h_P^\star))$ can be expressed as the normalized sum of $Z_{b\lceil \log_2\left(\frac{2}{\delta}\right)\rceil+1}, \ldots, Z_{b\lceil \log_2\left(\frac{2}{\delta}\right)\rceil+m}$ minus their expectation. Hence, by applying the one-sided Bernstein inequality we have

$$\mathbf{P}\left[\hat{er}_{\tilde{S}}(h) - \hat{er}_{\tilde{S}}(h_P^\star) - (er(h) - er(h_P^\star)) > \frac{\varepsilon}{4}\right] \leq \mathbf{P}\left[\frac{1}{m}\sum_{i=b\lceil \log_2\left(\frac{2}{\delta}\right)\rceil+1}^{b\lceil \log_2\left(\frac{2}{\delta}\right)\rceil+m}(Z_i - \mathbb{E}[Z_i]) > \frac{\varepsilon}{4}\right]$$

$$\leq \exp\left(-\frac{m\varepsilon^2/32}{\mathbb{E}[Z_i^2] + \varepsilon/12}\right).$$

---

[8]It is sufficient to find $n_0$ for which $n_0 \geq C \frac{\beta \log((1-2\beta)^2 n_0)}{(1-2\beta)\varepsilon}$, where $C$ is the absolute constant from 2. This works for $n_0 := C \frac{2\beta \log(1/\varepsilon)}{(1-2\beta)\varepsilon}$.

The next step is to upper bound the second moment of $Z_i$, which is in fact the probability of disagreement between $h_P^\star$ and $h$:

$$\mathbb{E}\left[Z_i^2\right] = \mathbb{E}\left[\left(\mathbb{1}_{\{h(X_i)\neq Y_i\}} - \mathbb{1}_{\left\{h_P^\star(X_i)\neq Y_i\right\}}\right)^2\right] = \mathbb{E}\left[\left(\mathbb{1}_{\left\{h(X_i)\neq h_P^\star(X_i)\right\}}\right)^2\right] = \mathbf{P}\left[h(X_i)\neq h_P^\star(X_i)\right]$$

Thus, from the definition of excess risk we have

$$er(h) - er(h_P^\star) = \mathbb{E}\left[\mathbb{1}_{\left\{h(X)\neq h_P^\star(X)\right\}}\left(1 - 2\mathbf{P}\left[h_P^\star(X)\neq Y|X\right]\right)\right] \geq (1-2\beta)\,\mathbf{P}\left[h(X_i)\neq h_P^\star(X_i)\right]$$

and we can deduce that

$$\mathbb{E}\left[Z_i^2\right] \leq \frac{\varepsilon}{2\left(1-2\beta\right)}$$

Putting it all together we get that

$$\mathbf{P}\left[\hat{er}_{\tilde{S}}(h) - \hat{er}_{\tilde{S}}(h_P^\star) > \frac{3\varepsilon}{4}\right] \leq \exp\left(-\frac{m\varepsilon^2/32}{\mathbb{E}[Z_i^2]+\varepsilon/12}\right) \leq e^{-\frac{m\varepsilon(1-2\beta)}{32}}$$

Now let us fix some $h \in A_{bad}$. Using multiplicative Chernoff bound we have

$$\mathbf{P}\left[\hat{er}_{\tilde{S}}(h) - \hat{er}_{\tilde{S}}(h_P^\star) < \frac{3\varepsilon}{4}\right] \leq \mathbf{P}\left[\hat{er}_{\tilde{S}}(h) - \hat{er}_{\tilde{S}}(h_P^\star) < \frac{3}{4}\left(er(h) - er(h_P^\star)\right)\right] \leq e^{-\frac{m\varepsilon}{32}}$$

Now, define the following event

$$B_2 = \left\{\exists h \in A_{good} : \hat{er}_{\tilde{S}}(h) - \hat{er}_{\tilde{S}}(h_P^\star) > \frac{3\varepsilon}{4}\right\} \cup \left\{\exists h \in A_{bad} : \hat{er}_{\tilde{S}}(h) - \hat{er}_{\tilde{S}}(h_P^\star) < \frac{3\varepsilon}{4}\right\}.$$

Then the probability of $B_2$ is at most

$$\left\lceil\log_2\left(\frac{2}{\delta}\right)\right\rceil e^{-\frac{m\varepsilon(1-2\beta)}{32}} \leq \left\lceil\log_2\left(\frac{2}{\delta}\right)\right\rceil\left(\frac{\delta}{2\lceil\log_2(2/\delta)\rceil}\right) \leq \frac{\delta}{2}$$

We conclude the proof by noting that the result holds by union bound over $B_1, B_2$, with probability of at most $\delta$.

## E. Proof of Universal Upper Bounds (Theorem 4)

This appendix section is dedicated to the proof of Theorem 4. As was discussed in section 4, we will use a single proof for both universal rates of $O(1/n)$ and $O(\log(n)/n)$. Our proof will share most of the structure as in the proof of the super-rates for universal agnostic learning[9], using Theorem 1 to allow the TERM algorithm to obtain a tighter bound for $\beta$-bounded distributions. Whilst this is straightforward for the case of a concept class with an infinite Littlestone tree but has no infinite VCL tree, the application for the linear rates case, that is, concept classes with infinite Littlestone tree, but no infinite star tree requires additional work. In their work, (Hanneke et al., 2024) have proven the existence of a learning rule based on Gale-Stewart star game that allows one to rule out star sets. Although they have proven that a sufficiently-large batch size can be learned that allows one to produce a "good" forbidden function with probability greater than $1/2$, their result holds only for data sequences coming from a realizable distribution. Fortunately, one may use the same argument for VC sets from the super-root proof of (Hanneke & Moran, 2026).

We start by citing the existing learning rules, starting with the pattern avoidance function $\text{SOA}_t^{VCL}$, which allows one to rule out VC sets (Bousquet et al., 2021). This is summarized well in (Hanneke & Moran, 2026)[Lemma 11]

---

[9]More specifically, as every $\beta$-bounded distribution is also Bayes-realizable (that is, $\inf_{h\in\mathcal{H}} er_P(h) = \inf_h er_P(h)$ where the infimum in the RHS is taken with respect to all measurable functions), our setting does not require the use of a universal Bayes-consistence learning algorithm of (Hanneke et al., 2021).

**Lemma 12** ((Bousquet et al., 2021)). *For any concept class $\mathcal{H}$ which does not shatter an infinite VCL tree, there exists a sequence $SOA_t^{VCL} : \times_{k=1}^{t-1} (\mathcal{X} \times \{0,1\})^k \times \mathcal{X}^t \to \{0,1\}^t$ of universally measurable functions such that, there does not exist an infinite sequence $\{S_t\}_{t \in \mathbb{N}}$ of finite sequences $S_t = \{(x_1^t, y_1^t), \ldots, (x_t^t, y_t^t)\} \in (\mathcal{X} \times \{0,1\})^t$ such that the sequence $S_1 \cup S_2 \cup \ldots$ is realizable with respect to $\mathcal{H}$ and*

$$\forall t \in \mathbb{N}, SOA_t^{VCL}(S_{<t}, (x_1^t, \ldots, x_t^t)) = (y_1^t, \ldots, y_t^t)$$

As we apply these learning rules onto random sequences, the exact batch size required to obtain certainty that we are not observing forbidden patterns is unknown and cannot be estimated. Fortunately, the following lemma asserts that there exists a distribution-dependent batch size that is sufficient to guarantee that the above occurs with constant probability.

**Lemma 13** ((Bousquet et al., 2021)). *Let $\mathcal{H}$ be any concept class which does not shatter an infinite VCL tree. For every $b \in \mathbb{N}$, there exists a universally measurable function $k_{VCL}^b : (\mathcal{X} \times \{0,1\})^b \to [b]$, and for every $k \in [b]$ there exists a universally measurable function $\mathcal{A}_{VCL}^{b,k} : (\mathcal{X} \times \{0,1\})^b \times \mathcal{X}^k \to \{0,1\}^k$ such that, for any distribution $P$ on $\mathcal{X} \times \{0,1\}$ which is realizable with respect to $\mathcal{H}$, $\forall \gamma \in (0,1)$, there exists $b_\gamma^* \in \mathbb{N}$ such that for every $b \geq b_\gamma^*$, $S_b \sim P^b$, and $\mathbf{k} = k_{VCL}^b(S_b)$, the following holds:*

$$\mathbf{P}\left[P^k\left[(x_1, y_1), \ldots, (x_\mathbf{k}, y_\mathbf{k}) : \mathcal{A}_{VCL}^{b,\mathbf{k}}(S_b, x_1, \ldots, x_\mathbf{k}) = (y_1, \ldots, y_\mathbf{k})\right] = 0\right] > \gamma$$

As we wish also to obtain linear universal rates, we are in need of a learning rule that allows us to rule out star sets. One such rule was made (Hanneke et al., 2024) using similar techniques to show the existence of exponential universal rates. The next two lemmas summarize their results, which parallel the two lemmas above.

**Lemma 14** ((Hanneke et al., 2024)). *For any concept class $\mathcal{H}$ which does not shatter an infinite star tree, there exists a sequence $SOA_t^{STAR} : \times_{k=1}^{t-1} (\mathcal{X} \times \{0,1\})^k \times \mathcal{X}^t \to \{0,1\}^t$ of universally measurable functions such that, there does not exist an infinite sequence $\{S_t\}_{t \in \mathbb{N}}$ of finite sequences $S_t = \{(x_1^t, y_1^t), \ldots, (x_t^t, y_t^t)\} \in (\mathcal{X} \times \{0,1\})^t$ such that the sequence $S_1 \cup S_2 \cup \ldots$ is realizable with respect to $\mathcal{H}$ and*

$$\forall t \in \mathbb{N}, SOA_t^{STAR}(S_{<t}, (x_1^t, \ldots, x_t^t)) = (y_1^t, \ldots, y_t^t)$$

**Lemma 15** ((Hanneke et al., 2024)). *Let $\mathcal{H}$ be any concept class which does not shatter an infinite star tree. For every $b \in \mathbb{N}$, there exists a universally measurable function $k_{STAR}^b : (\mathcal{X} \times \{0,1\})^b \to [b]$, and for every $k \in [b]$ there exists a universally measurable function $\mathcal{A}_{STAR}^{b,k} : (\mathcal{X} \times \{0,1\})^b \times \mathcal{X}^k \to \{0,1\}^k$ such that, for any distribution $P$ on $\mathcal{X} \times \{0,1\}$ which is realizable with respect to $\mathcal{H}$, $\forall \gamma \in (0,1)$, there exists $b_\gamma^* \in \mathbb{N}$ such that for every $b \geq b_\gamma^*$, $S_b \sim P^b$, and $\mathbf{k} = k_{STAR}^b(S_b)$, the following holds:*

$$\mathbf{P}\left[P^k\left[(x_1, y_1), \ldots, (x_\mathbf{k}, y_\mathbf{k}) : \mathcal{A}_{STAR}^{b,\mathbf{k}}(S_b, x_1, \ldots, x_\mathbf{k}) = (y_1, \ldots, y_\mathbf{k})\right] = 0\right] > \gamma$$

We emphasize that the learning algorithms we will present shortly, which are based on the applications of the universally measurable functions presented in Lemmas 12-15 are universally measurable: Our $O(\log(n)/n)$ learning algorithm is in fact identical to the learning algorithm $\hat{h}_n^1$ of (Hanneke & Moran, 2026), which was shown by them to be universally measurable. The same can be proven using similar arguments for our $O(1/n)$ learning rule. We refer the interested reader to (Hanneke & Moran, 2026)[Remark 13] for more information.

In Algorithm 1, our sample $S$ is split into three equal subsamples $S_1, S_2, S_3$, where the first two parts are used to estimate a sufficient batch size such that the pattern avoidance functions allows to produce a learning algorithm enable to obtain the required learning rate. This procedure was shown to be possible by (Bousquet et al., 2021) for realizable distributions, and was later extended to the agnostic setting by (Hanneke & Moran, 2026) by modifying the batch size selection criteria while using uniform convergence arguments. We next formulate the latter statement as a lemma, which requires us to introduce the construction of partial concept classes obtained from pattern avoidance functions. Starting with some definitions, we define $B_{b,i}^\mathbf{y} = ((X_{(i-1)b+1}, y_{(i-1)b+1}), \ldots, (X_{ib}, y_{ib}))$ for every $b \in [\lfloor n/3 \rfloor], i \in [\lfloor n/3b \rfloor]$, where $\mathbf{y} = (y_1, \ldots, y_b) \in \{0,1\}^{(X_1, \ldots, X_{\lfloor n/3 \rfloor})}$. For brevity, we also define $k_{b,i,\mathbf{y}}^{VCL} = k_{VCL}^b(B_{b,i}^\mathbf{y})$ and $f_{b,i,\mathbf{y}}^{VCL} = \mathcal{A}_{VCL}^{b,i,k_{b,\mathbf{y}}^{VCL}}(B_b^\mathbf{y})$. Using those, we define the function $G_{b,i,\mathbf{y}}^{VCL} : \cup_{t=1}^\infty (\mathcal{X} \times \{0,1\})^t \to \{0,1\}$, such that $\forall t \in \mathbb{N}$ and for every $(x_1, y_1), \ldots, (x_t, y_t) \in (\mathcal{X} \times \{0,1\})^t$, we have $G_{b,i,\mathbf{y}}^{VCL}(x_1, y_1, \ldots, x_t, y_t) = 1$ if: (a) $\forall 1 \leq i < j \leq t : x_i = x_j \Rightarrow y_i = y_j$. (b) For every distinct set of

$k_{b,i,\mathbf{y}}^{VCL}$ points $(x_{i_1}, \ldots, x_{i_{k_{b,i,\mathbf{y}}^{VCL}}}) \in \{x_1, \ldots, x_t\}$, we have $f_{b,i,\mathbf{y}}^{VCL}(x_{i_1}, \ldots, x_{i_{k_{b,i,\mathbf{y}}^{VCL}}}) \neq (y_{i_1}, \ldots, y_{i_{k_{b,i,\mathbf{y}}^{VCL}}})$. Next, for every $\cup_{t=1}^{\infty} (x_1, y_1, \ldots, x_t, y_t)$ we define the function

$$G_{b,i}^{VCL}(x_1, y_1, \ldots, x_t, y_t) = \bigvee_{\mathbf{y} \in \{0,1\}^{\lfloor n/3 \rfloor}} G_{b,i,\mathbf{y}}^{VCL}(x_1, y_1, \ldots, x_t, y_t).$$

Now, let $\mathcal{X}_{S_2}$ to be the set of all distinct points from $\mathcal{X}$ in $S_2$. For every $1 \leq b \leq \lfloor n/3 \rfloor$ and $1 \leq i \leq \lfloor n/3b \rfloor$, we define the partial concept class $\tilde{H}_{b,i}^{VCL}$ such that its projection onto $\mathcal{X}_{S_2}$ consists of all $\tilde{\mathbf{y}} = (0,1)^{(X_{\lfloor n/3 \rfloor + 1}, \ldots, X_{2\lfloor n/3 \rfloor})}$ satisfies $G_{b,i}^{VCL}(X_{\lfloor n/3 \rfloor + 1}, \tilde{y}_1, \ldots, X_{2\lfloor n/3 \rfloor}, \tilde{y}_{2\lfloor n/3 \rfloor}) = 1$. To make $\tilde{H}_{b,i}^{VCL}$ well-defined, we set every concept in $\tilde{H}_{b,i}^{VCL}$ to be undefined on $\mathcal{X} \setminus \mathcal{X}_{S_2}$. Finally, we define $E_{b,i}^{VCL}$ to the event that $h_P^{\star}$ labelings are not included in the projection of $\tilde{H}_{b,i}^{VCL}$ on $\mathcal{X}_{S_2}$. More formally,

$$E_{b,i}^{VCL} := \left\{ \left( h_P^{\star}(X_{\lfloor n/3 \rfloor + 1}), \ldots, h_P^{\star}(X_{2\lfloor n/3 \rfloor}) \right) \notin \tilde{H}_{b^*,i}^{VCL} \left[ X_{\lfloor n/3 \rfloor + 1}, \ldots, X_{2\lfloor n/3 \rfloor} \right] \right\}.$$

We are now ready to state the lemma.

**Lemma 16** ((Hanneke & Moran, 2026)). *There exists a universally measurable function $\hat{b}_n = \hat{b}_n$ $(X_1, Y_1, \ldots, X_{2\lfloor n/3 \rfloor}, Y_{2\lfloor n/3 \rfloor})$, whose definition does not depend on $P$, so that the following holds. Given $b^*$ such that $\mathbf{P}\left[ E_{b^*,i}^{VCL} \right] \leq 1/10$ for some fixed $i$, then there exists $C > 0$ independent of $n$ (but depending on $P, b^*$) so that*

$$\mathbf{P}\left[ \hat{b}_n \in \mathcal{T}_{good} \right] \geq 1 - \frac{C}{n}$$

*where*

$$\mathcal{T}_{good} := \left\{ 1 \leq b \leq b^* : \mathbf{P}\left[ E_{b,i}^{VCL} \right] \leq \frac{3}{10} \right\}$$

*Remark* E.1. We emphasize that the existence $b^*$ is guaranteed by the following argument: The event $E_{b,i}^{VCL}$ is equivalent to the event $\left\{ G_{b,i}^{VCL}\left( X_{\lfloor n/3 \rfloor + 1}, h_P^{\star}(X_{\lfloor n/3 \rfloor + 1}), \ldots, X_{2\lfloor n/3 \rfloor}, h_P^{\star}(X_{2\lfloor n/3 \rfloor}) \right) = 0 \right\}$, which in turn is contained in the event $\left\{ G_{b,i,\mathbf{y}^*}^{VCL}\left( X_{\lfloor n/3 \rfloor + 1}, h_P^{\star}(X_{\lfloor n/3 \rfloor + 1}), \ldots, X_{2\lfloor n/3 \rfloor}, h_P^{\star}(X_{2\lfloor n/3 \rfloor}) \right) = 0 \right\}$, where the latter can be interpreted as the event where the VC pattern avoidance function learned over a i.i.d data sequence sampled from the realizable distribution $P^*$ over $\mathcal{X} \times \{0,1\}$ for which $\mathbf{P}\left[ Y = h_P^{\star}(X) | X \right] = 1$ for $(X, Y) \sim P^*$. Hence, by Lemma 13 choosing $b^* := b_{9/10}^*$ satisfies the demand made in Lemma 16.

We are now left with showing that a sufficiently good batch size could be estimated for the sake of the star pattern avoidance function in our setting. (Hanneke et al., 2024) has proved this for realizable distributions by paralleling the same arguments of (Bousquet et al., 2021) for VCL pattern avoidance functions. Fortunately enough, this batch size estimation in our case is immediate by Lemma 16. Indeed, batch size assessment is made by comparing the performance of partial concept classes based on different batch sizes, where the partial concept classes constructed by VC pattern avoidance functions is shown to have VC-dimension of at most $b^* 2^{b^*}$.[10] Following the same procedure for star pattern avoidance functions results in partial concept classes with star number at most $b^* 2^{b^*}$, thus implying these partial concept classes have indeed a VC dimension at most $b^* 2^{b^*}$. For completeness, we prove this also for star pattern avoidance functions in the next lemma, where we adapt the notation made for Lemma 16 such that $\left( k_{b,i,\mathbf{y}}^{STAR}, f_{b,i,\mathbf{y}}^{STAR}, G_{b,i,\mathbf{y}}^{STAR}, E_{b,i}^{STAR} \right)$ will keep their respective former meaning, with the sole difference that they are defined with respect to the tuple $\left( k_{STAR}^b, \mathcal{A}_{STAR}^{b,k} \right)$.

**Lemma 17** (Adaptation of Lemma 16 from (Hanneke & Moran, 2026)). *There exists a universally measurable function $\hat{b}_n = \hat{b}_n(X_1, Y_1, \ldots, X_{2\lfloor n/3 \rfloor}, Y_{2\lfloor n/3 \rfloor})$, whose definition does not depend on $P$, so that the following holds. Given $b^*$ such that $\mathbf{P}\left[ E_{b^*,i}^{STAR} \right] \leq 1/10$, there exists $C > 0$ independent of $n$ (but depending on $P, b^*$) so that*

$$\mathbf{P}\left[ \hat{b}_n \in \mathcal{T}_{good}^{STAR} \right] \geq 1 - \frac{C}{n}$$

---

[10]Conditionally on $S_1$ and the support of $S_2$, see (Hanneke & Moran, 2026)[Theorem 24])

*where*

$$\mathcal{T}_{good}^{STAR} := \left\{ 1 \leq b \leq b^* : \mathbf{P}\left[E_{b,i}^{STAR}\right] \leq \frac{3}{10} \right\}$$

**Proof:** We start by defining our estimation process. For the remainder of the proof, we omit the upper-script $STAR$ from all related quantities, replacing it with the index $i$, to keep notation more clear and concise. For each $(b, i)$ we define $\hat{e}_b^i$ to be the minimal empirical error of $\tilde{H}_b^i$ on $\mathcal{X}_{S_2}$. That is,

$$\hat{e}_b^i := \frac{1}{\lfloor n/3 \rfloor} \min_{\tilde{h} \in \tilde{H}_b^i} \sum_{i=\lfloor n/3 \rfloor + 1}^{2\lfloor n/3 \rfloor} \mathbb{1}_{\left\{\tilde{h}(X_i) \neq Y_i\right\}}$$

Next, we define the following estimate

$$\hat{\psi}_b^i := \mathbb{1}_{\left\{\hat{e}_b^i \leq \hat{e}_{b'}^{i'} + \sqrt{\frac{2b' 2^{b'} \log(n^3)}{\lfloor n/3 \rfloor}} \text{ for every } b' > b, \, i' \in [\lfloor n/3b' \rfloor] \right\}}$$

$\hat{\psi}_b^i$ can be interpreted as a benchmark for whether $\tilde{H}_b^i$, the $i$-th partial concept class learned on a $b$-sized batch, is sufficiently good as other partial concept classes learned using pattern avoidance function learned over a larger batch-size.

To this end, we select our estimate to be

$$\hat{b}_n := \inf\left\{ b \in [\lfloor n/3 \rfloor] : \frac{1}{\lfloor n/3b \rfloor} \sum_{i \in [\lfloor n/3b \rfloor]} \hat{\psi}_b^i \geq \frac{8}{10} \right\}$$

where we assume that $\inf \emptyset = \infty$.

We introduce additional notation to streamline the proof. Given a classifier $h : \mathcal{X}_{S_2} \to \{0, 1\}$, we denote $er_{P|\mathcal{X}_{S_2}}(h) := \mathbf{P}\left[h(X) \neq Y | X \in \mathcal{X}_{S_2}\right]$. Let $E_{UC}$ be the event that every classifier in all the partial concept classes constructed over $\mathcal{X}_{S_2}$ has its empirical error on $S_2$ converges to $er_{P|\mathcal{X}_{S_2}}(h)$. More formally,

$$E_{UC} = \bigcap_{\substack{\forall b \in [\lfloor n/3 \rfloor] \\ \forall i \in [\lfloor n/3b \rfloor]}} \left\{ \forall \tilde{h} \in \tilde{H}_b^i : \hat{er}_{S_2}(\tilde{h}) - er_{P|\mathcal{X}_{S_2}}(\tilde{h}) \leq \sqrt{\frac{b2^b \log\left(\lfloor n/3b \rfloor^3\right)}{\lfloor n/3 \rfloor}} \right\}$$

We claim that $\mathbf{P}\left[\overline{E_{UC}}\right] = 1/n$. To see this, we first condition on $S_1$ and on $\mathcal{X}_{S_2}$, allowing us to fix all partial concept classes over $\mathcal{X}_{S_2}$, where each partial concept class $\tilde{H}_b^i$ has VC dimension of at most $b2^b$ for all $i \in [\lfloor n/3b \rfloor], b \in [\lfloor n/3 \rfloor]$. In addition, conditioning on $\mathcal{X}_{S_2}$ allows us to treat the projections of all partial concept classes $\tilde{H}_b^i$ as a total concept classes over $X_{S_2} \times \{0, 1\}$, so that uniform convergence could be applied. Thus, for a fixed $b, i$ we have that with probability of at least $1 - \delta$, the following holds

$$\forall \tilde{h} \in \tilde{H}_b^i : \left|\hat{er}_{S_2}(\tilde{h}) - er_{P|\mathcal{X}_{S_2}}(\tilde{h})\right| \leq \sqrt{\frac{b2^b + \log\left(\frac{1}{\delta}\right)}{\lfloor n/3 \rfloor}}$$

Choosing $\delta = n^{-3}$, the above can be transformed into a guarantee on the entire set $\left\{\tilde{H}_b^i\right\}_{\substack{b \in [\lfloor n/3 \rfloor] \\ i \in [\lfloor n/3b \rfloor]}}$ using the union bound.

Since we have at most $n^2$ such partial concept classes, we arrive at the conclusion that

$$\mathbf{P}\left[\left|\hat{er}_{S_2}(\tilde{h}) - er_{P|\mathcal{X}_{S_2}}(\tilde{h})\right| > \sqrt{\frac{2b2^b \log(n^3)}{\lfloor n/3 \rfloor}} \text{ for some } \tilde{h} \in \tilde{H}_b^i, \, b \leq n, \, i \in \lfloor n/3b \rfloor\right] \leq \frac{1}{n}$$

Hence, by the above and the law of total probability we may write

$$\mathbf{P}\left[\hat{b}_n \notin \mathcal{T}_{good}\right] \leq \mathbf{P}\left[\hat{b}_n > b^*\right] + \mathbf{P}\left[\hat{b}_n < \inf \mathcal{T}_{good}\right]$$

Starting with bounding the probability of the event that our estimate to exceed $b^*$, this event is contained by the event $\left\{\frac{1}{\lfloor n/3b^*\rfloor}\sum_{i\in[\lfloor n/3b^*\rfloor]}\hat{\psi}^i_{b^*} < \frac{8}{10}\right\}$, hence it is sufficient to bound the probability of the latter.

Let us define the event $E_{(b^*)\text{-GOOD}} = \left\{\frac{1}{\lfloor n/3b^*\rfloor}\sum_{i\in[\lfloor n/3b^*\rfloor]}\mathbb{1}_{\{E^i_{b^*}\}} \leq \frac{1}{5}\right\}$, by the law of total probability we have

$$\mathbf{P}\left[\frac{1}{\lfloor n/3b^*\rfloor}\sum_{i\in[\lfloor n/3b^*\rfloor]}\hat{\psi}^i_{b^*} < \frac{8}{10}\right]$$

$$= \mathbf{P}\left[\frac{1}{\lfloor n/3b^*\rfloor}\sum_{i\in[\lfloor n/3b^*\rfloor]}\hat{\psi}^i_{b^*} < \frac{8}{10}, \overline{E_{(b^*)\text{-GOOD}}}\right] + \mathbf{P}\left[\frac{1}{\lfloor n/3b^*\rfloor}\sum_{i\in[\lfloor n/3b^*\rfloor]}\hat{\psi}^i_{b^*} < \frac{8}{10}, E_{(b^*)\text{-GOOD}}\right]$$

$$\leq \mathbf{P}\left[\overline{E_{(b^*)\text{-GOOD}}}\right] + \mathbf{P}\left[\frac{1}{\lfloor n/3b^*\rfloor}\sum_{i\in[\lfloor n/3b^*\rfloor]}\hat{\psi}^i_{b^*} < \frac{8}{10}, E_{(b^*)\text{-GOOD}}\right]$$

By Remark E.1 the *complementary event* of $E_{(b^*)\text{-GOOD}}$ is contained in the event $\left\{\frac{1}{\lfloor n/3b^*\rfloor}\sum_{i\in[\lfloor n/3b^*\rfloor]}\xi_{b^*,i} > \frac{1}{5}\right\}$, where $\xi_{b^*,i}$ is the indicator function for the event that the $i$-th forbidden pattern avoidance function learned over a $b^*$-sized batch from $S_1$ has a non-zero probability to observe $h^\star_P$ labelings as a forbidden set, that is,

$$P^{*k}\left[(X'_1, h^\star_P(X'_1)), \ldots, (X'_{b^*}, h^\star_P X'_{b^*}) : \mathcal{A}^{b^*,i,k^{STAR}_{b,i,\mathbf{y}^*}}_{STAR}(B^{\mathbf{y}^*}_{b,i}, X'_1, \ldots, X'_{k^{STAR}_{b,i,\mathbf{y}^*}}) = \left(h^\star_P(X'_1), \ldots, h^\star_P(X'_{k^{STAR}_{b,i,\mathbf{y}^*}})\right)\right] = 0.$$

By Lemma 15, the probability of $\xi_{b^*,i}$ is at most $1/10$, and thus by Hoeffding inequality we obtain the bound

$$\mathbf{P}\left[\overline{E_{(b^*)\text{-GOOD}}}\right] \leq \mathbf{P}\left[\frac{1}{\lfloor n/3b^*\rfloor}\sum_{i\in[\lfloor n/3b^*\rfloor]}\xi_{b,i} > \frac{1}{5}\right] \leq e^{-\frac{2\lfloor n/3b^*\rfloor}{100}}$$

Moving on to the second term, using again the law of total probability using the event $E_{UC}$, we obtain

$$\mathbf{P}\left[\frac{1}{\lfloor n/3b^*\rfloor}\sum_{i\in[\lfloor n/3b^*\rfloor]}\hat{\psi}^i_{b^*} < \frac{8}{10}, E_{(b^*)\text{-GOOD}}\right]$$

$$\leq \mathbf{P}\left[\overline{E_{UC}}\right] + \mathbf{P}\left[\frac{1}{\lfloor n/3b^*\rfloor}\sum_{i\in[\lfloor n/3b^*\rfloor]}\hat{\psi}^i_{b^*} < \frac{8}{10}, E_{(b^*)\text{-GOOD}}, E_{UC}\right]$$

$$\leq \frac{1}{n} + \mathbf{P}\left[\frac{1}{\lfloor n/3b^*\rfloor}\sum_{i\in[\lfloor n/3b^*\rfloor]}\hat{\psi}^i_{b^*} < \frac{8}{10} \,\middle|\, E_{(b^*)\text{-GOOD}}, E_{UC}\right]$$

To analyze the probability of the latter event, it is sufficient to look at the subset of $\left\{\tilde{H}^i_{b^*}\right\}_{i\in[\lfloor n/3b^*\rfloor]}$ that contains $h^\star_P$ labelings over $\mathcal{X}_{S_2}$ (as we condition on $E_{(b^*)\text{-GOOD}}$, there are at least $\lfloor 8/10\rfloor$ which satisfies this). Let $I_{(b^*)\text{-GOOD}} \subset \lfloor n/3b^*\rfloor$ be the set of indices for which $\left(h^\star_P(X_{\lfloor n/3\rfloor+1}), \ldots, h^\star_P(X_{2\lfloor n/3\rfloor})\right) \in \tilde{H}^i_{b^*}\left[X_{\lfloor n/3\rfloor+1}, \ldots, X_{2\lfloor n/3\rfloor}\right]$, in order for the event $\left\{\frac{1}{\lfloor n/3b^*\rfloor}\sum_{i\in[\lfloor n/3b^*\rfloor]}\hat{\psi}^i_{b^*} < \frac{8}{10}\right\}$ to occur, there must be an index $i \in \left[I_{(b^*)\text{-GOOD}}\right]$ such that the following occurs

$$\left\{\hat{e}^i_{b^*} > \hat{e}^{i'}_{b'} + 2\sqrt{\frac{2b'2^{b'}\log(n^3)}{\lfloor n/3\rfloor}} \text{ for some } b' > b^*, \, i' \in \lfloor n/3b'\rfloor\right\}.$$

However, since we condition on the event $E_{UC}$, we have that for any $\hat{e}_{b'}^{i'}, \hat{e}_{b*}^i$, the following holds

$$\hat{e}_{b'}^{i'} \geq er_{P|\mathcal{X}_{S_2}}(h_P^\star) - \sqrt{\frac{2b'2^{b'}\log(n^3)}{\lfloor n/3 \rfloor}}, \qquad \hat{e}_{b*}^i \leq er_{P|\mathcal{X}_{S_2}}(h_P^\star) + \sqrt{\frac{2b*2^{b*}\log(n^3)}{\lfloor n/3 \rfloor}}$$

This is because $er_{P|\mathcal{X}_{S_2}}(h) \geq er_{P|\mathcal{X}_{S_2}}(h_P^\star)$ and $\min_{h\in\hat{H}_{b*}^i} \hat{er}_{S_2}(h) \leq \hat{er}_{S_2}(h_P^\star)$ hold for every $h$ in the projection $\tilde{H}_{b*}^i [X_{\lfloor n/3 \rfloor + 1}, \ldots, X_{2\lfloor n/3 \rfloor}]$. Thus, we reach the conclusion that the probability of the event $\left\{ \frac{1}{\lfloor n/3b* \rfloor} \sum_{i\in[\lfloor n/3b* \rfloor]} \hat{\psi}_{b*}^i < \frac{8}{10} \right\}$ given the events $E_{(b*)\text{-GOOD}}$ and $E_{UC}$ is zero, since

$$\hat{e}_{b*}^i \leq er_{P|\mathcal{X}_{S_2}}(h_P^\star) + \sqrt{\frac{2b*2^{b*}\log(n^3)}{\lfloor n/3 \rfloor}} \leq \hat{e}_{b'}^{i'} + 2\sqrt{\frac{2b'2^{b'}\log(n^3)}{\lfloor n/3 \rfloor}}$$

Next, we move on to bound the probability of the event where $\hat{b}_n$ returns a too small batch size. Note that by union bound, it is sufficient to bound the probability of the event $\left\{ \hat{b} = b' \right\}$ for some arbitrary $b' < \inf \mathcal{T}_{good}$, since

$$\mathbf{P}\left[ \hat{b}_n < \inf \mathcal{T}_{good} \right] = \mathbf{P}\left[ \exists b < \inf \mathcal{T}_{good} \text{ such that } \frac{1}{\lfloor n/3b \rfloor} \sum_{i\in[\lfloor n/3b \rfloor]} \psi_b^i \geq \frac{8}{10} \right]$$

$$\leq b^* \mathbf{P}\left[ \frac{1}{\lfloor n/3b' \rfloor} \sum_{i'\in[\lfloor n/3b' \rfloor]} \psi_{b'}^{i'} \geq \frac{8}{10} \right].$$

Fix some arbitrary $b' < \inf \mathcal{T}_{good}$ such that we have $\mathbf{P}\left[ E_{b',i}^{STAR} \right] > \frac{3}{10}$ and define the conditional excess error of the partial concept class of $\tilde{H}_b^i$ (with respect to the probability distribution $P|\mathcal{X}_{S_2}$, that is, conditioned on the event that $X \in \mathcal{X}_{S_2}$) to be

$$\triangle er_{P|\mathcal{X}_{S_2}}(\tilde{H}_b^i) := \min_{\tilde{h}\in\tilde{H}_b^i} er_{P|\mathcal{X}_{S_2}}(\tilde{h}) - er_{P|\mathcal{X}_{S_2}}(h_P^\star).$$

Now, as the event $\left\{ E_{b',i}^{STAR} \right\}$ is equivalent to the event in which the concept class $\tilde{H}_{b'}^i$ has excess error greater than zero with respect to the distribution $P|\mathcal{X}_{S_2}$, by continuity of the measure there exists a $P$-dependent $\varepsilon > 0$ such that the following hold:

$$\mathbf{P}\left[ \triangle er_{P|\mathcal{X}_{S_2}}(\tilde{H}_{b'}^i) > \varepsilon \right] > \frac{1}{4}$$

Therefore, the probability that at most fifth of the $\tilde{H}_{b'}^i$ have excess error greater than $\varepsilon$, which denote $E_{\triangle er(b')>\varepsilon}$, can be bounded by Hoeffding

$$\mathbf{P}\left[ \frac{1}{\lfloor n/3b' \rfloor} \sum_{i'\in[\lfloor n/3b' \rfloor]} \mathbb{1}_{\left\{ \triangle er_{P|\mathcal{X}_{S_2}}(\tilde{H}_{b'}^i) > \varepsilon \right\}} \leq \frac{1}{5} \right] \leq e^{-\frac{\lfloor n/3b' \rfloor}{200}} \leq e^{-\frac{\lfloor n/3b* \rfloor}{200}}$$

Thus, using the law of total probability using the events $E_{UC}, E_{\triangle er(b')>\varepsilon}$ in a similar manner as before, we have that

$$\mathbf{P}\left[ \frac{1}{\lfloor n/3b' \rfloor} \sum_{i'\in[\lfloor n/3b' \rfloor]} \psi_{b'}^{i'} \geq \frac{8}{10} \right] \leq e^{-\frac{\lfloor n/3b* \rfloor}{200}} + \mathbf{P}\left[ \frac{1}{\lfloor n/3b' \rfloor} \sum_{i'\in[\lfloor n/3b' \rfloor]} \psi_{b'}^{i'} \geq \frac{8}{10} \,\bigg|\, E_{\triangle er(b')>\varepsilon} \right]$$

$$\leq e^{-\frac{\lfloor n/3b* \rfloor}{200}} + \frac{1}{n} + \mathbf{P}\left[ \frac{1}{\lfloor n/3b' \rfloor} \sum_{i'\in[\lfloor n/3b' \rfloor]} \psi_{b'}^{i'} \geq \frac{8}{10} \,\bigg|\, E_{UC}, E_{\triangle er(b')>\varepsilon} \right]$$

Using previous arguments, note that the latter probability can be bounded as follows:

$$\mathbf{P}\left[\frac{1}{\lfloor n/3b'\rfloor}\sum_{i'\in[\lfloor n/3b'\rfloor]}\psi_{b'}^{i'}\geq\frac{8}{10}\;\Big|\;E_{UC},E_{\triangle er(b')>\varepsilon}\right]\leq\mathbf{P}\left[\exists i'\in\lfloor n/3b'\rfloor:\triangle er_{P|\mathcal{X}_{S_2}}(\tilde{H}_{b'}^{i'})>\varepsilon\right.$$

$$\text{such that }\hat{e}_{b'}^{i'}\leq\hat{e}_{b^*}^{i}+2\sqrt{\frac{2b^*2^{b^*}\log(n^3)}{\lfloor n/3\rfloor}}\text{ for all }i\in\lfloor n/3b^*\rfloor\;\Big|\;E_{UC},E_{\triangle er(b')>\varepsilon}\right]$$

Using the condition over the event of uniform convergence $E_{UC}$, we have that

$$\hat{e}_{b'}^{i'}\geq er_{P|\mathcal{X}_{S_2}}(h_P^\star)+\varepsilon-\sqrt{\frac{2b'2^{b'}\log(n^3)}{\lfloor n/3\rfloor}},\qquad\qquad\hat{e}_{b^*}^{i}\leq er_{P|\mathcal{X}_{S_2}}(h_P^\star)+\sqrt{\frac{2b^*2^{b^*}\log(n^3)}{\lfloor n/3\rfloor}}$$

As the above implies $\hat{e}_{b'}^{i'}>\hat{e}_{b^*}^{i}+\varepsilon-2\sqrt{\frac{2b^*2^{b^*}\log(n^3)}{\lfloor n/3\rfloor}}$, we can conclude that for a sufficiently large $n$ such that $\varepsilon>2\sqrt{\frac{2b^*2^{b^*}\log(n^3)}{\lfloor n/3\rfloor}}$, and that the probability $\mathbf{P}\left[\frac{1}{\lfloor n/3b'\rfloor}\sum_{i'\in[\lfloor n/3b'\rfloor]}\psi_{b'}^{i'}\geq\frac{8}{10}\;\Big|\;E_{UC},E_{\triangle er(b')>\varepsilon}\right]$ is zero. $\qquad\square$

Finally, we move on to the proof of Theorem 4. The proof is done only for the $O(1/n)$ rate, as it is identical for the $O(\log(n)/n)$ rate.

**Proof:**

We adopt the notations from the proof of Lemma 17. The sample is split into three equal parts, $S_i=S_{(i-1)\lfloor n/3\rfloor+1,i\lfloor n/3\rfloor}$ where $i\in[3]$. We use the first part to learn sets of pattern avoidance functions for each batch size $b\in[\lfloor n/3\rfloor]$. The second part is used to compute $\hat{b}_n$ in the manner described in Lemma 17. Using the last part of the sample, we use the selected set of pattern avoidance functions $\left\{G_{\hat{b}_n,i}^{STAR}:i\in\left[\left\lfloor n/3\hat{b}_n\right\rfloor\right]\right\}$ to construct partial concept classes $\left\{\hat{H}_{\hat{b}_n}^i:i\in\left[\left\lfloor n/3\hat{b}_n\right\rfloor\right]\right\}$ over $X_{S_3}=X_{2\lfloor n/3\rfloor+1},\ldots,X_{3\lfloor n/3\rfloor+1},X$. Finally, Algorithm 1 is used to produce a collection of predictors $\left\{\hat{h}_n^i:=\mathcal{A}\left(S_3,\hat{H}_{\hat{b}_n}^i,X\right):i\in\left[\left\lfloor n/3\hat{b}_n\right\rfloor\right]\right\}$, and the prediction on $X$ is determined by the majority voting among them.

We start with the following fact that the excess risk of some $h:\mathcal{X}\to\{0,1\}$ can be expressed as

$$er(h)-er(h_P^\star)=\mathbb{E}\left[\mathbb{1}_{\{h(X)\neq h_P^\star(X)\}}(1-2\mathbf{P}[h_P^\star(X)\neq Y|X])\right]$$

with the expectation taken with respect to $P_X$.

Using this, we have

$$\mathbb{E}[er(\hat{h}_n)-er(h_P^\star)]=\mathbb{E}\left[\mathbb{1}_{\{\hat{h}_n(X)\neq h_P^\star(X)\}}(1-2\mathbf{P}[h_P^\star(X)\neq Y|X])\right]$$

By Lemma 17 we have that $\mathbf{P}\left[\hat{b}_n\notin\mathcal{T}_{good}\right]=O(1/n)$. Thus

$$\mathbb{E}\left[\mathbb{1}_{\{\hat{h}_n(X)\neq h_P^\star(X)\}}(1-2\mathbf{P}[h_P^\star(X)\neq Y|X])\right]$$

$$\leq C/n+\mathbb{E}\left[\mathbb{1}_{\{\hat{h}_n(X)\neq h_P^\star(X)\}}\mathbb{1}_{\{\hat{b}_n\in\mathcal{T}_{good}\}}(1-2\mathbf{P}[h_P^\star(X)\neq Y|X])\right]$$

Let us denote $J$ the set of all indices in $\left[\left\lfloor n/3\hat{b}_n\right\rfloor\right]$ such that $h_P^\star$ labelings appear in the projection $\hat{H}_{\hat{b}_n}^j\left[X_{2\lfloor n/3\rfloor+1},\ldots,X_{3\lfloor n/3\rfloor}\right]$. On the event that $\hat{b}_n\in\mathcal{T}_{good}$, we have that

$$\mathbf{P}\left[(h_P^\star(X_{2\lfloor n/3\rfloor+1}),\ldots,h_P^\star(X_{3\lfloor n/3\rfloor}))\notin\hat{H}_{\hat{b}_n}\left[X_{2\lfloor n/3\rfloor+1},\ldots,X_{3\lfloor n/3\rfloor}\right]\right]\leq\frac{3}{10}$$

Therefore, by Hoeffding we have that $\mathbf{P}\left[|J| < \frac{6}{10} \,|\, \hat{b}_n \in \mathcal{T}_{good}\right] \leq e^{-\lfloor n/3\hat{b}_n\rfloor/50}$, which in turn implies that

$$\mathbb{E}\left[\mathbb{1}_{\{\hat{h}_n(X) \neq h_P^\star(X)\}} \mathbb{1}_{\{\hat{b}_n \in \mathcal{T}_{good}\}} \left(1 - 2\mathbf{P}\left[h_P^\star(X) \neq Y|X\right]\right)\right] \leq e^{-\lfloor n/3\hat{b}_n\rfloor/50} +$$

$$\mathbb{E}\left[\mathbb{1}_{\{\hat{h}_n(X) \neq h_P^\star(X)\}} \mathbb{1}_{\{\hat{b}_n \in \mathcal{T}_{good}\}} \mathbb{1}_{\{|J| \geq \frac{6}{10}\}} \left(1 - 2\mathbf{P}\left[h_P^\star(X) \neq Y|X\right]\right)\right]$$

On the event that $\hat{h}_n(X) \neq h_P^\star(X)$ we have that at least $\frac{1}{6}$ of $J$, which are at least $\frac{1}{10}$ of all $\left[\lfloor n/3\hat{b}_n\rfloor\right]$, do not agree with $h_P^\star$. More formally

$$\left\{\hat{h}_n(X) \neq h_P^\star(X)\right\} \subseteq \left\{\frac{1}{|J|} \sum_{j \in J} \mathbb{1}_{\{\hat{h}_n^j(X) \neq h_P^\star(X)\}} \geq \frac{1}{6}\right\}$$

Hence, the expectation can be bounded as follows:

$$\mathbb{E}\left[\mathbb{1}_{\{\hat{h}_n(X) \neq h_P^\star(X)\}} \mathbb{1}_{\{\hat{b}_n \in \mathcal{T}_{good}\}} \mathbb{1}_{\{|J| \geq \frac{6}{10}\}} \left(1 - 2\mathbf{P}\left[h_P^\star(X) \neq Y|X\right]\right)\right]$$

$$\leq \mathbb{E}\left[\mathbb{1}_{\left\{\frac{1}{|J|} \sum_{j \in [J]} \mathbb{1}_{\{\hat{h}_n^j(X) \neq h_P^\star(X)\}} \geq \frac{1}{6}\right\}} \mathbb{1}_{\{\hat{b}_n \in \mathcal{T}_{good}\}} \mathbb{1}_{\{|J| \geq \frac{6}{10}\}} \left(1 - 2\mathbf{P}\left[h_P^\star(X) \neq Y|X\right]\right)\right]$$

$$\leq 6\mathbb{E}\left[\left(\frac{1}{|J|} \sum_{j \in [J]} \mathbb{1}_{\{\hat{h}_n^j(X) \neq h_P^\star(X)\}}\right) \mathbb{1}_{\{\hat{b}_n \in \mathcal{T}_{good}\}} \mathbb{1}_{\{|J| \geq \frac{6}{10}\}} \left(1 - 2\mathbf{P}\left[h_P^\star(X) \neq Y|X\right]\right)\right]$$

where the last transition is due to the identity $T \geq \gamma \mathbb{1}_{\{T \geq \gamma\}}$ for the non-negative random variable $T := \frac{1}{|J|} \sum_{j \in [J]} \mathbb{1}_{\{\hat{h}_n^j(X) \neq h_P^\star(X)\}}$.

Finally, on the event of $\left\{\hat{b}_n \in \mathcal{T}_{good}\,,\, |J| \geq \frac{6}{10}\right\}$ we condition on $X_{S_3}$ to use the expectation bound from Theorem 1,

$$\mathbb{E}\left[\frac{1}{|J|} \sum_{j \in [J]} \mathbb{1}_{\{\hat{h}_j^n(X) \neq h_P^\star(X)\}} \left(1 - 2\mathbf{P}\left[h_P^\star(X) \neq Y|X\right]\right)\right]$$

$$\leq \frac{1}{|J|} \sum_{j \in [J]} \mathbb{E}\left[\mathbb{1}_{\{\hat{h}_j^n(X) \neq h_P^\star(X)\}} \left(1 - 2\mathbf{P}\left[h_P^\star(X) \neq Y|X\right]\right)\right]$$

$$\leq \frac{1}{|J|} \sum_{j \in [J]} \mathbb{E}\left[er(\hat{h}_n^j)\right] - er(h_P^\star) \leq C\left(\frac{b^* 2^{b^*} \log\left(\frac{b^* 2^{b^*}}{1 - 2\beta}\right)}{(1 - 2\beta)n}\right)$$

where we have used the fact that $d \leq \mathfrak{s} \leq b^* 2^{b^*}$ for any member of the partial concept classes set $\left\{\hat{H}_{\hat{b}_n}^i : i \in \left[\lfloor n/3\hat{b}_n\rfloor\right]\right\}$. Finally, collecting all the terms, we get

$$\mathbb{E}[er(\hat{h}_n) - er(h_P^\star)] \leq C/n + e^{-2\lfloor n/3\hat{b}_n\rfloor/100} + C\left(\frac{b^* 2^{b^*} \log\left(\frac{b^* 2^{b^*}}{1 - 2\beta}\right)}{(1 - 2\beta)n}\right) = O(1/n)$$

$\square$

## F. Slower than Linear is not Faster than log(n)/n

In this section we prove the lower bound for the case where there exists an infinite star tree for the (total) concept class $\mathcal{H}$.

Before proving this result, we cite a lemma which serves as a fundamental for the uniform lower bound (Hanneke, 2016, Lemma 22). The setting for the lemma is the following. Fix $\zeta \in (0, 1], \beta \in [0, 1/2)$, and $k \in \mathbb{N}$ with $k \leq \min\{1/\zeta, |\mathcal{X}| - 1\}$. Let $\mathcal{X}_k = \{x_1, ..., x_{k+1}\}$ be a set of $k + 1$ distinct elements of $\mathcal{X}$, and $h_0, h_1, ..., h_k$ be concepts defined over $\mathcal{X}$ as in Definition 8. Let $P_{k,\zeta}$ be a distribution over $\mathcal{X}$ with $P_{k,\zeta}[x_i] = \zeta$ for each $i \in [k]$, and $P_{k,\zeta}[x_{k+1}] = 1 - \zeta k$, and for each $t \in [k]$, let $P_{k,\zeta,t}$ denote a distribution over $\mathcal{X} \times \{0, 1\}$ with marginal distribution $P_{k,\zeta}$ over $\mathcal{X}$ such that, for $(X, Y) \sim P_{k,\zeta,t}, \mathbf{P}[Y = h_t(X)|X = x_i] = 1 - \beta$ for every $i \in [k]$, while $\mathbf{P}[Y = h_t(X)|X = x_{k+1}] = 1$.

**Lemma 18.** *For $k, \zeta, \beta$ as above with $k \geq 96e$, for any $\delta \in (0, 1/4)$, for any learning algorithm $\hat{h}_n$, and any $n \in \mathbb{N}$ with*

$$n < \frac{3\beta \log(k/96)}{16\zeta(1 - 2\beta)^2}$$

*there exists a $t \in [k]$ such that if $(X, Y) \sim P_{k,\zeta,t}$, then with probability greater than $\delta$,*

$$er(\hat{h}_n) - \inf_{h \in \mathcal{H}} er(h) \geq \frac{\zeta(1 - 2\beta)}{2}$$

In particular, if we choose $\zeta = 1/k$ and some arbitrary $\delta \in (0, 1/4)$, for $(X, Y) \sim P_{k,\zeta,t}$ we have

$$\mathbb{E}\left[er(\hat{h}_n) - \inf_{h \in \mathcal{H}} er(h)\right] \geq \frac{\delta(1 - 2\beta)}{2k}$$

Lemma 18 is built upon a lower bound by (Raginsky & Rakhlin, 2011), which is based on information-theoretic techniques. Specifically, (Raginsky & Rakhlin, 2011) used $f$-divergences to compare between two probability measures on $\mathcal{X} \times \{0, 1\}$, where $\mathcal{X}$ is taken to be countable: a $\beta$-bounded probability measure over the first $k$ points, and a dummy measure. While these two measures shared the same marginal distribution over $\mathcal{X}$, they share the same conditional distribution on the labels only outside the first $k$ points, that is, on the point $x_{k+1}$ (in case such point was needed, where $1 - \zeta k > 0$). As identical conditional distributions do not reflect on the overall divergence [11], any bias in $[0, 1/2]$ could be applied on the $k + 1$ point. More generally, both measures could be extend outside the $k$ points, so long they share the same conditional distribution at every point on that region. We will use this argument to build the distribution used in the lower bound of Theorem 5.

**Proof:** We start by fixing an arbitrary learning algorithm $\hat{h}_n$, $\beta \in (0, 1/2), \delta \in (0, 1/4)$ and an infinite tree for $\mathcal{H}$, $t = \{(x_{\mathbf{u}}, y_{\mathbf{u}}) \in (\mathcal{X} \times \{0, 1\})^{k+2} : \mathbf{u} \in \prod_{j=2}^{k+1}[j], 0 \leq k < \infty\}$.

As it is sufficient to prove the above for infinitely many $n$, we define the sequences $\{k_i\}_{i=1}^{\infty}, \{n_i\}_{i=1}^{\infty}$ and the set of probabilities $p_1, p_2, ...$ to be:

$$k_1 := 256, \ k_i := \inf\left\{k > k_{i-1} : \frac{1}{k} \leq 2^{-i} \min_{j<i} \frac{2^j}{k_j n_j}\right\}$$

$$n_i := \left\lfloor \frac{\beta k_i^2 \log(k_i/96)}{1366(1 - 2\beta)^2} \right\rfloor, \ p_k := C \frac{\mathbb{1}_{\{k \in \{k_i\}_{i=1}^{\infty}\}}}{k},$$

where $C = \left(\sum_{k \geq 1} p_k\right)^{-1} \leq 256$. Note that $k_{i+1} > 2k_i$ for all $i \in \mathbb{N}$.

We will now fix an infinite branch in the tree to apply Lemma 18 recursively. Let $\mathbf{w} = (w_1 w_2 ...) \in \prod_{j \geq 1}[j]$, we set each $w_j$ inductively as follows. If $j \in \{k_i\}_{i=1}^{\infty}$, we define $w_j$ to be

$$w_j = \left\{t_j \in [j] : \mathbf{P}\left[er(\hat{h}_n) - er(h_P^\star) \geq \frac{1 - 2\beta}{2j}\right] \geq \delta, (X, Y) \sim P_{j,\zeta_j,t_j}\right\}$$

---

[11]See (Raginsky & Rakhlin, 2011, Lemma 3) for more details

where $P_{j,\zeta_j,t_j}$ is a member of the set of distributions $\left\{ P_{j,\zeta_j,t}, t \in [j] \,\Big|\, \zeta_j := \frac{p_{k_j}}{k\sum_{l\leq j} p_{k_l}} \right\}$, which satisfy the following properties:

- The marginal support of each distribution is limited to the set $\{x \in x_{\mathbf{w}\leq j'} : j' \leq j\}$.

- For each $j' < j$, the conditional distribution over $\{x_{\mathbf{w}\leq j'}\}$ is identical for all distributions in the set, that is, for every $t, t' \in [j]$ we have $\mathbf{P}_{P_{j,\zeta_j,t}}[\cdot|X \in x_{\mathbf{w}\leq j'}] = \mathbf{P}_{P_{j,\zeta_j,t'}}[\cdot|X \in x_{\mathbf{w}\leq j'}]$.

- Given that $x \in x_{\mathbf{w}\leq j}$, the conditional distribution of $P_{j,\zeta_j,t}$ is defined as:

$$\mathbf{P}_{P_{j,\zeta_j,t}}\left[X = x^i_{\mathbf{w}\leq j}|X \in x_{\mathbf{w}\leq j}\right] = \frac{1}{j} \text{ for every } i \in [j]$$

$$\mathbf{P}_{P_{j,\zeta_j,t}}\left[Y = y^i_{\mathbf{w}\leq j}|X = x^i_{\mathbf{w}\leq j}\right] = \begin{cases} 1-\beta & i \neq t \\ \beta & i = t \end{cases}$$

Otherwise, we set $w_j = 1$ arbitrarily. Note that by Lemma 18 each $w_j$ is well-defined.

Next, we define the distribution $P_w$, starting with its marginal distribution over $\mathcal{X}$,

$$P_w\left[x^i_{\mathbf{w}\leq k}, \cdot\right] = \frac{p_k}{k} \text{ for every } i \in [k], k \in \mathbb{N}.$$

Before defining the conditional distribution of $P_w$, recall from the definition of the star tree that for every $k < \infty$, there exist $h_k \in \mathcal{H}$ such that for every $k' \leq k$ we have that

$$h_k(x^{w_{k'}}_{\mathbf{w}\leq k'}) = 1 - y^{w_{k'}}_{\mathbf{w}\leq k'}, \text{ and for all } j \in [k'] \setminus \{w_{k'}\} : h_k(x^j_{\mathbf{w}\leq k}) = y^j_{\mathbf{w}\leq k}.$$

For each $k \in \{k_i\}_{i=1}^\infty$ and $j \in [w_k]$, we define

$$P_w\left[h_k(X) = Y|X = x^i_{\mathbf{w}\leq k}\right] = 1 - \beta \text{ for every } 1 \leq i \leq k$$

To verify that $P_w$ is a $\beta$-bounded distribution, observe that for $i \geq 1$, there exists a $h_{k_i}$ such that the marginal probability of $\{x : P_w[h_{k_i}(X) \neq Y|x] > \beta\}$ is at most $\sum_{l > k_i} p_l$ which can be bounded as follows:

$$\sum_{l > k_i} p_l = \sum_{j > i} p_{k_j} \leq \frac{C}{k_i} \sum_{j > i} 2^{i-j} = \frac{C}{k_i}$$

Hence, the existence of the sequence of $h_{k_i}$ establishes that $P_w$ is indeed a $\beta$-bounded distribution.

Now, let $X, Y, X_1, Y_1, ..., X_n, Y_n$ be independent random variables drawn from $P_w$, let us define the following random variables

$$T = \{k \in \mathbb{N} : X \in x_{\mathbf{w}\leq k}\}, \qquad\qquad T_i = \{k \in \mathbb{N} : X_i \in x_{\mathbf{w}\leq k}\}$$

Recall that

$$\mathbb{E}\left[er(\hat{h}_n) - \inf_{h \in \mathcal{H}} er(h)\right] = \mathbb{E}\left[\mathbb{1}_{\{\hat{h}_n(X) \neq h^\star_P(X)\}}(1 - 2\mathbf{P}[h^\star_P(X) \neq Y|X])\right] = (1 - 2\beta)\mathbf{P}\left[\hat{h}_n(X) \neq h^\star_P(X)\right]$$

Observe that for every $i \geq 1$ we have

$$\mathbf{P}\left[h_{n_i}(X) \neq h^\star_P(X)\right] \geq \mathbf{P}\left[h_{n_i}(X) \neq h^\star_P(X), T, T_1, ..., T_{n_i} \leq k_i\right]$$

$$= \mathbf{P}\left[h_{n_i}(X) \neq h^\star_P(X)|T, T_1, ..., T_{n_i} \leq k_i\right]\left(1 - \sum_{l > k_i} p_l\right)^{n_i+1}$$

Note that $1 - \sum_{l>k_i} p_l \geq 1 - \sum_{l>k_1} p_l = \frac{C}{256}$. In addition, we have

$$\sum_{l>k} p_l = \sum_{j>i} p_{k_j} = \sum_{j>i} \frac{C}{k_j} = C \sum_{j>i} \inf\left\{ k > k_{j-1} : \frac{1}{k} \leq 2^{-j} \min_{j'<j} \frac{2^{j'}}{k_{j'} n_{j'}} \right\} \leq C \sum_{j>i} 2^{-j} \frac{2^i}{k_i n_i} \leq \frac{C}{n_i} \sum_{j>i} 2^{i-j} = \frac{C}{n_i}$$

Putting it together we get

$$\left( 1 - \sum_{l>k_i} p_l \right)^{n_i+1} \geq \frac{C}{256} \left( 1 - \frac{C}{n_i} \right)^{n_i} \geq \frac{Ce^{-C}}{256}$$

Thus, we have that the event $\{T, T_1, ..., T_{n_i} \leq k_i\}$ occurs with probability at least $\frac{Ce^{-C}}{256}$. Now, let us consider the conditional probability $\mathbf{P}\left[ h_{n_i}(X) \neq h_P^\star(X) | T, T_1, ..., T_{n_i} \leq k_i \right]$. By conditioning on the event $\{T, T_1, ..., T_{n_i} \leq k_i\}$ we have that both the data set and the test point are sampled from depth at most $k_i$, that is, from the conditional distribution $P_w\left(\cdot | T, T_1, ..., T_n \leq k_i\right)$, which is identical to $P_{k_i, \zeta_{k_i}, w_{k_i}}$, where $\zeta_{k_i} := \frac{p_{k_i}}{k_i \sum_{j \leq i} p_{k_j}} = \frac{1}{k_i^2 \sum_{j \leq i} k_j^{-1}}$. Observing that $\zeta_{k_i} = \Theta\left(k_i^{-2}\right)$[12] we ensure $n_i < \frac{3\beta \log(k_i/96)}{16 \zeta_{k_i}(1-2\beta)^2}$, thus satisfying the conditions of Lemma 18. Therefore we get

$$(1-2\beta) \mathbf{P}\left[ \hat{h}_n(X) \neq h_P^\star(X) | T, T_1, ..., T_n \leq k_i \right] = \mathbb{E}\left[ er(\hat{h}_n) - \inf_{h \in \mathcal{H}} er(h) | T, T_1, ..., T_n \leq k_i \right] \geq \frac{\zeta_{k_i} \delta (1-2\beta)}{2}$$

Next, for sufficiently large $i$ and using the fact that $n_i = \Theta\left(k_i^2 \log(k_i)\right)$, we have

$$\frac{\zeta_{k_i} \delta(1-2\beta)}{2} \geq \frac{\delta(1-2\beta)}{2k_i^2} \gtrsim \frac{\log(k_i)}{n_i} \gtrsim \frac{\log(n_i)}{n_i}$$

where the last inequality follows from

$$\log n_i \leq \log\left( ck_i^2 \log(k_i)\right) \leq \log\left( ck_i^3\right) \leq 3c \log(k_i)$$

As this bound holds for infinitely many $n$, our proof is concluded. $\qquad\square$

---

[12]To see this, note that $1/256 = 1/k_1 \leq \sum_{j \leq i} k_j^{-1} \leq 1$.

