# OpenReview forum: "Learning Partial Concept Classes and Universal Rates Under Massart Noise"
_ICML.cc/2026/Conference — ICML 2026 regular_

### Official Review · Reviewer_AC5T · 2026-03-12

**Soundness:** 2
**Presentation:** 3
**Significance:** 3
**Originality:** 3
**Overall Recommendation:** 3
**Confidence:** 4

**Summary:**

This paper studies two extensions of classical statistical learning theory to the Massart ($\beta$-bounded) noise setting. First, it sharpens the excess-error guarantee of the transductive empirical risk minimization (TERM) algorithm of Hanneke and Moran for partial concept classes under weak $\beta$-bounded noise, using offset Rademacher complexity and local metric entropy. The paper explicitly positions this as a sharpening of the previous $o\left(\mathrm{VC}(H)/n\right)$ type in-expectation guarantee for partial classes.

Second, it derives improved PAC sample-complexity bounds for partial concept classes under Massart noise from this TERM bound. Third, and most importantly, it claims a full characterization of universal learning rates under bounded noise: $e^{-n}$, exactly $e^{-o(n)}$, exactly $1/n$, exactly $\log n / n$, and arbitrarily slow, depending on whether the class is finite or admits infinite Littlestone, star, or VCL trees. The paper presents this three-part contribution explicitly in the introduction.

The strongest part of the paper is the universal-rate story. The claim that the Massart setting yields two intermediate universal rates, $1/n$ and $\log n / n$, connecting universal learnability to the classical uniform complexity measures, is mathematically interesting and non-obvious. The paper also appears to introduce a genuinely new technical ingredient in adapting the batch-size estimation machinery from the VCL-pattern setting to the star-pattern setting, which it uses to establish the universal linear-rate regime.

**Compliance With Llm Reviewing Policy:**

Affirmed.

**Key Questions For Authors:**

1) The universal-rate theorem states the $1/n$ regime as exact. Please make the lower-bound argument for this case explicit in the paper, including the exact reduction or citation you rely on.

2) Please promote the projection-based monotonicity/transfer argument used in Theorem 1 into a formal lemma, so the reduction from partial classes to projected realizable classes is fully explicit.

3) For Theorem 2, please clarify in the theorem statement or discussion that the approximation-error interpretation of the paper’s excess-error notion requires additional conditions, as stated in the background discussion.

4) Please clarify whether the logarithmic gap in the partial-class sample-complexity bounds is merely a consequence of the confidence-boosting argument, or whether it reflects a deeper barrier.

**Limitations:**

The main limitation is that the paper's partial-concept PAC bounds are not yet tight: even the authors note that standard bounded-noise PAC theory leaves room for improvement. A second limitation is that the interpretation of the excess-error guarantees under weak $\beta$-bounded noise relies on additional conditions such as countable support or finite VC dimension. Finally, while the universal-rate pentachotomy is the paper's strongest and most complete contribution, the broader framing sometimes overstates completeness relative to the still-unmatched sample-complexity results.

**Strengths And Weaknesses:**

$\textbf{Strengths}$

The paper's contribution structure is clear and ambitious. It addresses partial concept classes and universal learning under Massart noise, two settings that the introduction argues had not yet been combined in this way for bounded-noise binary classification.

Theorem 1 is technically meaningful. The paper does not merely restate prior TERM guarantees; it explicitly sharpens the previous in-expectation bound for partial classes by importing offset-Rademacher/local-entropy techniques into the weak-$\beta$ setting.

The universal-rate result is the highlight. The manuscript explicitly states a five-regime Massart pentachotomy and contrasts it with prior agnostic and ERM universal-rate classifications. This is the main reason the paper is interesting.

The local-metric-entropy machinery is not merely invoked superficially. The appendix defines local empirical packing and local metric entropy for realizable projections, then applies a localization lemma and a VC/star-number entropy bound to control the offset sub-Gaussian process.


$\textbf{Weaknesses}$

My main concerns are about completeness and theorem presentation, not about lack of novelty.

$\textbf{1) The ``exact $1/n$'' lower bound is not surfaced clearly enough in the theorem flow.}$

 The paper states a full characterization of universal rates and presents the $1/n$ regime as exact, but in the visible structure, the explicit new lower bound in the universal section is $\log n / n$, while the arbitrarily slow case is attributed to prior work.
This does not necessarily mean the $\Omega(1/n)$ lower bound is false or absent; it likely follows from known bounded-noise lower bounds once infinite VC dimension is inferred. But if that is the intended route, it should be stated explicitly. As written, the paper's ``full characterization'' claim is stronger than the clarity of the exposition supporting it.

$\textbf{2) The key projection step in Theorem 1 should be formalized more cleanly.}$

 The appendix defines local metric entropy directly on realizable projections and states that these are the relevant objects for the argument. So the transfer from partial classes to projected total classes is not absent.
However, because this step is central, the paper would be stronger if it were stated explicitly as a lemma: namely, that the projected realizable class inherits the relevant VC/star-number control needed to apply the local-entropy bound. At present, this point is present, but more in prose than in theorem-grade form.

$\textbf{3) Theorem 2 is useful, but it is not tight and should not be framed on the same footing as the universal characterization.}$

 The paper's introduction and conclusion present a broad, complete-investigation message. However, the excess-error/sample-complexity part is an improvement, not a closed characterization. In particular, the paper itself notes that its excess-error notion aligns with the more standard approximation-error interpretation under additional conditions such as countable support or finite VC dimension.
This does not invalidate Theorem 2, but it does mean the theorem's operational interpretation is narrower than the broad statement suggests.

$\textbf{4) Dependence on recent unpublished scaffolding is substantial.}$

 The paper openly builds on Hanneke-Moran for the TERM algorithm and the agnostic universal-rate story. That is acceptable, but it increases the verification burden.

$\textbf{Soundness}$

The technical route for Theorem 1 is coherent: the paper defines excess error under bounded noise, explains the weak/strong distinction, and then builds the sharpened TERM bound via projection-based localization and local metric entropy.

My concern is not that I found a concrete contradiction, but rather that some central proof dependencies are not elevated to the level of explicit, self-contained lemmas, which matters in a theory paper making a "complete characterization" claim.


$\textbf{Originality}$

The significance is good. The universal-rate pentachotomy is the genuine contribution here. The emergence of the $1/n$ and $\log n / n$ regimes under Massart noise is the part most likely to matter to the theory community. The partial-class TERM/sample-complexity results are useful, but secondary.

The originality is also good, but not radical. The paper is partly a bridge-building work: TERM, offset-Rademacher methods, local entropy, and pattern-avoidance machinery are imported. The novelty lies in how these are combined in the partial-class/Massart setting, especially the star-pattern batch-size adaptation and the resulting five-rate universal picture.

---

> ### Author Rebuttal · Authors · 2026-03-31
>
> We sincerely thank the reviewer for their careful evaluation of our paper. We appreciate that the reviewer thoroughly grasped the contribution structure, recognizing the universal-rate pentachotomy under Massart noise as mathematically interesting and acknowledging the novelty of extending offset-Rademacher and local entropy techniques to partial concept classes.
>
> ### 1. Missing explicit 1/n lower bound argument
> We agree this critical bounding argument was incorrectly condensed in the submitted version. We will re-add the missing text: *"If H has an infinite Littlestone tree, then H is not beta-universally learnable at a rate faster than 1/n."*
>
> This follows from realizable universal learning bounds [Theorem 4.6 in Bousquet et al., 2021], since if H cannot be learned faster than 1/n without noise, the same lower limit strictly applies when introducing the structurally harder Massart noise assumption. We will make this reduction completely explicit in the camera-ready version.
>
> ### 2. Key projection step in Theorem 1 formalized
> We will formally clarify the step bounding local entropy via subset projections. Note that the non-increasing nature of combinatorial complexities (VC / star numbers) upon subset restriction is an elementary attribute of their definitions. To improve clarity without duplicating the symmetrical mechanism used in Theorem 2's lower bounds, we are adding a dedicated footnote breaking down the projection step: pointing out that projection to a finite set cannot increase the VC/star numbers, and restricting to realizable concepts only further reduces them.
>
> ### 3. Theorem 2 and approximation-error interpretation
> We thank the reviewer for this precise and fair critique, and we accept your guidance on adjusting the terminology regarding our sample complexity bounds:
>
> *   **Removal of the word 'optimal':** We agree that Theorem 2 does not provide a closed, tight characterization in the same vein as our universal rates. The use of the word "optimal" was indeed an editorial oversight, and we have reframed the contribution as a "considerable improvement" for partial concept classes under Massart noise.
> *   **The scale of the improvement:** While the bounds are not completely tight, Theorem 2 remains highly significant. It successfully improves the sample complexity dependence from the O(1/ε²) rate known for the agnostic setting to rates scaling as O(1/ε) and O(log(1/ε)/ε) under Massart noise. We have revised the introduction to highlight this specific polynomial improvement.
> *   **Operational interpretation and finite VC dimension:** The equivalence of our excess-error definition with the approximation error under a finite VC dimension is not a restriction in practice. If the VC dimension is infinite, the sample complexity is inherently infinite under *both* the Bayes-relative expected error and the approximation-error notions. Therefore, our excess-error notion naturally aligns with the approximation-error interpretation for all classes that are actually PAC learnable.
>
> ### 4. Logarithmic gap in partial-concept sample complexity
> The logarithmic penalty is an artifact of the confidence-boosting step we utilized (e.g., median-of-means procedures). It does not reflect a deep statistical barrier for transductive Massart limits. Theoretically, bridging this gap requires translating the expected guarantee of our TERM algorithm to a leave-one-out bound, followed by the rigorous application of recent transductive martingale arguments introduced by Dughmi et al. (2024). This path would yield entirely tight high-probability bounds without boosting gaps. We will reflect this pathway more clearly in our conclusion.
>
> ### 5. Dependence on recent unpublished scaffolding
> First, we respectfully point out that the referenced agnostic universal learning paper has been uploaded to arXiv (see https://arxiv.org/abs/2601.20961), and we therefore engage with its conceptual framework directly.
>
> Regarding concerns about the magnitude of this dependence, we would like to strongly emphasize that all of the results continuously derived in our paper are entirely mathematically standalone. Every aspect of our technical derivations—for the refined TERM algorithm bounds, the PAC sample complexity for partial concept classes, and the novel intermediate universal rates—is proven explicitly and independently within the mechanics of the Massart noise setting.
>
> We only rely on the aforementioned paper to reference the fast (near-exponential) universal rates. We use these conceptually as a structural baseline to show that, *together* with our novel intermediate rates (log(n)/n and 1/sqrt(n)), we successfully complete the full universal characterization pentachotomy for Massart noise. For any remaining concerns regarding novelty delineation, we courteously refer the reviewer to the rebuttal for reviewer 1jAr.

---

> > ### Author Rebuttal · Reviewer_AC5T · 2026-04-03
> >
> > I thank the authors for the thoughtful rebuttal. The response improves the presentation and addresses some concerns, especially by clarifying the intended $1/n$ lower-bound route and by softening the claim around Theorem 2. However, my main concerns are only partially resolved. In particular, the projection-based transfer step in Theorem 1 remains underformalized, the current version still does not present all exact-rate claims as cleanly as the paper’s “full characterization” framing suggests, and Theorem 2 remains non-tight in the submitted version. Because these remaining concerns pertain to core aspects of the paper and are not easily resolved in a short rebuttal, my overall recommendation remains unchanged.

---

### Official Review · Reviewer_T9Ek · 2026-03-12

**Soundness:** 4
**Presentation:** 3
**Significance:** 4
**Originality:** 3
**Overall Recommendation:** 4
**Confidence:** 3

**Summary:**

This paper extends the Massart noise condition to two modern directions in statistical learning theory: partial concept classes and universal learning. Because standard Empirical Risk Minimization (ERM) is known to fail for partial concept classes, the authors instead analyze a transductive learning approach. The central technical contribution is a refined analysis of the TERM algorithm under Massart noise, leading to sharper excess error bounds, using tools such as offset Rademacher complexity and local metric entropy. Using this result, the authors derive two applications:
1. Improved PAC sample complexity bounds for learning partial concept classes under Massart noise, achieving rates of order \(1/\epsilon\) (up to logarithmic factors) instead of the \(1/\epsilon^2\) dependence that arises in the agnostic setting.
2. A characterization of universal learning rates under strong Massart noise, establishing a pentachotomy of possible asymptotic rates, including  \(\log(n)/n\), which does not appear either in the realizable setting or the agnostic setting.

**Compliance With Llm Reviewing Policy:**

Affirmed.

**Final Justification:**

I maintain my positive score.

----

## Update during Reviewer discussion period

I missed quite a few things in my review that Reviewer AC5T pointed out. I find that these issues are more substantial than what I understood at first glance. As a result, I have lowered my score from 5 to 4 for the paper.

**Key Questions For Authors:**

A natural direction for future work would be to extend the analysis to more general noise conditions, such as Tsybakov’s margin condition. Since Massart noise is a special case of Tsybakov noise, and the techniques used here (local entropy and offset Rademacher complexity) are commonly used to derive fast rates under margin conditions, it would be interesting to understand whether similar results can be obtained for partial concept classes and universal learning under Tsybakov noise. Do the authors have any thought on this topic?

### A few typos

1. line 146: emprical process theory
2. line 162-163: We emphasize that our algoritms rely on universally measurable functions imploded by... (also, should be citet instead of cite) .
3. line 423 second column: rates with a complementy lower bound

**Limitations:**

Given that the work is purely theoretical and develops new statistical learning theory results (analysis of learning rates under Massart noise), there are no obvious direct societal risks.

**Strengths And Weaknesses:**

## Strengths

1. The paper extends the Massart noise model, which is an important intermediate regime between realizable and agnostic settings, to both partial concept classes and universal learning. Since these frameworks had previously been analyzed mainly in the realizable and agnostic cases, this work helps complete the understanding of learnability across the standard noise regimes.

2. The core technical contribution is a refined analysis of the Transductive Empirical Risk Minimization (TERM) algorithm in the presence of bounded noise. The proof combines localization tools such as offset Rademacher complexity and local metric entropy with the structural challenges posed by partial concept classes and transductive learning. This extension of Massart-noise techniques to the partial-class setting appears technically delicate, and is an important addition to the statistical learning literature in its own right.

3. Using the TERM analysis as a building block, the paper derives a characterization of universal learning rates under bounded noise. The resulting pentachotomy of rates, including the appearance of a \(\log(n)/n\) regime, reveals a new intermediate statistical behavior that does not arise in previously studied realizable or agnostic universal-learning settings.

---

## Weaknesses

1. While the analysis of TERM under Massart noise is technically interesting, the algorithmic framework itself is not new. The main results follow largely by plugging this refined analysis into existing frameworks. As a result, the novelty of the work lies primarily in the analysis rather than in new algorithmic ideas.

2. The introduction occasionally suggests that classical PAC learning cannot accommodate assumptions such as margin conditions or structured data, which is not entirely accurate. These assumptions have been extensively studied within statistical learning theory. The contribution is better viewed as extending existing analyses to new frameworks rather than addressing fundamental limitations of PAC theory itself. See lines 9-14 (second column) for an example.

---

> ### Author Rebuttal · Authors · 2026-03-31
>
> We thank the reviewer for their careful reading and thoughtful feedback. We are glad to see that the reviewer recognizes the technical contributions of our work, particularly the refined analysis of the TERM algorithm under Massart noise and the characterization of universal learning rates. We also appreciate the constructive criticism regarding the framing of our contributions, which we will address in detail below.
>
> ### Weakness 1: Algorithmic novelty vs. analysis
> We agree with the reviewer that our primary contribution is analytical rather than proposing a fundamentally new algorithmic framework. However, we would like to emphasize that in statistical learning theory, deriving tight bounds and understanding the fundamental statistical limits of natural algorithms under challenging noise conditions is often the central goal.
>
> While the transductive ERM (TERM) framework itself is known, proving that it achieves optimal fast rates under Massart noise for partial concept classes is highly non-trivial. Bypassing the O(1/ε²) agnostic barrier to achieve O(1/ε) rates required a highly delicate synthesis of localization tools (such as offset Rademacher complexity and local metric entropy) tailored precisely to the structural challenges of transductive learning. We believe this novel, rigorous analysis not only fills an important gap in the literature, but also provides essential mathematical machinery that future work can build upon. We will make sure to state clearly in the introduction that our main novelty is theoretical and analytical.
>
> ### Weakness 2: Clarifying the scope of classical PAC learning
> We thank the reviewer for the insightful observation regarding the framing of margin conditions and structured data. We agree that these have been extensively studied within the broader context of statistical learning theory (SLT) via data-dependent bounds and Rademacher complexities (e.g., Shawe-Taylor et al., 1998).
>
> Our discussion in the introduction was intended to address a narrower, technical distinction regarding the *distribution-free* PAC framework (Valiant, 1984). As noted in the introduction of *A Theory of PAC Learnability of Partial Concept Classes* (Alon et al., 2021), classical margin-based analyses often deviate from this strict framework because the VC dimension of the underlying total concept classes may be infinite or scale with the ambient dimension. By employing the partial concept class framework, we can capture these assumptions (margins or low-dimensional manifolds) as combinatorial constraints that satisfy a purely distribution-free PAC guarantee.
>
> We will add an accompanying footnote to better reflect this nuance. Specifically, we will reframe the contribution as extending the *purely combinatorial, distribution-free PAC analysis* to these structured settings, explicitly acknowledging that they have been handled by other frameworks within SLT. This ensures our novelty is correctly positioned as a structural extension rather than a claim of solving an entirely unstudied problem.
>
> ### Extending the analysis to more general noise conditions
> We thank the reviewer for this insightful suggestion. Extending our framework to accommodate more general noise conditions is indeed a natural and promising avenue for future research.
>
> If one were to pursue a generalization of our results, it would likely be most advantageous to consider the (B,β)-Bernstein class condition, which formally generalizes Tsybakov’s margin condition and has become the more widely studied and utilized framework in contemporary learning theory.
>
> We emphasize that even under Tsybakov's margin condition, optimal uniform rates for total concept classes are only known up to a log factor gap between upper and lower bounds, so one would presumably first need to resolve this uniform question, before proceeding to the study of either universal rates or partial concept classes.
> We do note however that fast universal rates under these generalized noise conditions are immediate, as these extremes are already governed by the Massart and agnostic behaviors.
>
> Finally, to our knowledge, it remains an open question whether learning under the Bernstein condition can directly benefit from the offset Rademacher process analysis, similar to the machinery we utilized for Massart noise. Resolving this question will certainly benefit the analysis of the TERM under this noise model.
>
> We will happily add a discussion detailing this promising future direction to the conclusion of our paper.
>
> ### Typos and Minor Corrections
> We thank the reviewer for careful reading. All of the typos will be corrected in the camera-ready version.

---

> > ### Author Rebuttal · Reviewer_T9Ek · 2026-04-01
> >
> > I maintain my positive score.

---

### Official Review · Reviewer_uqNB · 2026-03-14

**Soundness:** 4
**Presentation:** 3
**Significance:** 3
**Originality:** 3
**Overall Recommendation:** 5
**Confidence:** 3

**Summary:**

This work studies PAC learning of partial concept classes under Massart noise and derives tighter sample complexity bounds for this setting. Furthermore, the authors characterize the universal learning rate of concept classes under Massart noise. Notably, their analysis makes use of the transductive empirical risk minimization (TERM) algorithm introduced by Hanneke and Moran (2026).

**Compliance With Llm Reviewing Policy:**

Affirmed.

**Key Questions For Authors:**

No questions

**Strengths And Weaknesses:**

Overall, I believe the community will find this paper interesting. The paper studies an important theoretical problem and develops new results for PAC learning of partial concept classes under Massart noise. The presentation is clear, the structure of the paper is well organized, and the arguments are generally easy to follow. I also found the exposition of the main ideas accessible despite the technical nature of the results. Therefore, I recommend acceptance.


Some minor issues:
- Line 377: The text says *“infinite Littlestone tree is identical.”* I believe this was intended to refer to an "infinite VCL tree".
- Line 707: The expression  $\sum_{i=1}^n \mathbb1_{\{g(x_i) \neq g^{\prime}(x_i)\}} > \varepsilon$, likely should be  $\sum_{i=1}^n \mathbb1_{\{g(x_i) \neq g^{\prime}(x_i)\}} > n\varepsilon$.
- Line 949: I believe $\mathfrak{s}$ should appear instead of $k$.
- Line 1115: *“for”* should be capitalized as *“For”*.

---

> ### Author Rebuttal · Authors · 2026-03-30
>
> We sincerely thank the reviewer for their positive feedback, their recommendation for acceptance, and for finding our presentation clear and accessible despite the technical nature of the results.
>
> ### Responses to Minor Issues and Corrections
> We also thank the reviewer for their careful reading and for catching these typos. We address them below and will correct them in the camera-ready version:
>
> **Line 377 (VCL tree):**
> Indeed, the reviewer is correct. The sketch is equivalent to the case where no infinite VCL tree exists for $\mathcal{H}$. This will be fixed.
>
> **Line 707 (Packing number distance):**
> Regarding the expression $\sum_{i \in [n]}\mathbb{I}[g(x_i) \ne g'(x_i)] > \epsilon$, we would like to clarify that the definition of packing numbers we use from Zhivotovskiy and Hanneke (2016) is taken with respect to the *unnormalized* Hamming distance. Please see the beginning of Section 4 in that reference for this definition. Therefore, the expression is correct as written.
>
> **Line 949 (Variables in summation):**
> The reviewer is perfectly correct; $k$ should be changed to $\mathfrak{s}$. We also point out that this fix will be carried out for line 947 as well, and $k$ should be changed to $k'$ in the expression on line 946.
>
> **Line 1115 (Capitalization):**
> The reviewer is correct. "for" will be capitalized as "For".

---

> > ### Author Rebuttal · Reviewer_uqNB · 2026-04-04
> >
> > I have no further questions.

---

### Official Review · Reviewer_1jAr · 2026-03-15

**Soundness:** 2
**Presentation:** 2
**Significance:** 3
**Originality:** 2
**Overall Recommendation:** 4
**Confidence:** 4

**Summary:**

This paper studies binary classification under the Massart (β-bounded) noise model beyond the classical PAC framework, focusing on two modern extensions: partial concept classes and universal learning rates. The authors (i) sharpen the excess error bound of a transductive ERM (TERM)-type algorithm (following Hanneke & Moran, 2026) under Massart noise using offset Rademacher techniques and local metric entropy (Zhivotovskiy & Hanneke, 2018), (ii) leverage this to derive tighter (apparently optimal-up-to-logs) sample complexity bounds for PAC learning of partial concept classes under a weak Massart condition, and (iii) complete a pentachotomy characterization of universal rates under Massart noise for total concept classes, yielding regimes with rates e−n, e−o(n), 1/n, log(n)/n, and arbitrarily slow.

**Compliance With Llm Reviewing Policy:**

Affirmed.

**Key Questions For Authors:**

1.Novelty delineation: Which proof components are substantially new relative to Hanneke & Moran (2026) and Zhivotovskiy & Hanneke (2018), versus re-packaging/adaptation? A sharper map would help assess originality.

2.Main-text clarity: Can you add a higher-level roadmap explaining how TERM + exchangeability reduces to controlling a localized empirical process, and where Massart-specific improvements enter?

3.Definition subtleties: Please clearly state, in the main text (not only appendix), under what conditions your Bayes-relative excess error is equivalent to the approximation-error notion used for partial concept classes.

**Limitations:**

yes

**Strengths And Weaknesses:**

Strengths.

Clear and meaningful gap filled. Massart noise is well-studied in classical PAC learning, but is substantially less developed in the universal-learning and partial-concept frameworks; the paper addresses this gap directly.

Technically well-motivated approach. Combining TERM-style transductive analysis (exchangeability) with offset-Rademacher/local-entropy machinery is a natural and credible route to Massart-optimal bounds.

Results “match the right shape.” The partial-concept sample complexity bounds align structurally with known Massart PAC bounds (dominant d/(ϵ(1−2β))-type behavior plus a βlog(⋅) refinement), suggesting near-optimality.

Complete universal-rate taxonomy under Massart noise. The pentachotomy is a clean, high-level contribution that clarifies the landscape of distribution-dependent rates in this intermediate noise regime.

Weaknesses.

Heavy reliance on very recent/under-review foundations. Key building blocks (notably Hanneke & Moran, 2026 and other recent universal-learning extensions) are essential to the framework. The paper should more explicitly delineate what is genuinely new versus inherited, and ensure that any crucial lemmas are fully stated/proved (at least in the appendix) for independent verification.

Presentation is steep for a broad ML audience. Universal measurability / Gale–Stewart game arguments and “pattern avoidance” constructions are hard to parse without strong scaffolding. Even for a theory-heavy venue, the main text would benefit from clearer roadmaps and intuition.

Subtleties in excess-error definitions under weak vs. strong Massart conditions should be surfaced more prominently. The equivalence between Bayes-relative excess error and inf_(ℎ∈𝐻) relative excess error depends on additional conditions (e.g., finite VC dimension or countable support), which are important for interpreting the sample complexity claims.

---

> ### Author Rebuttal · Authors · 2026-03-31
>
> We thank the reviewer for the opportunity to provide a sharper map of our original contributions relative to the foundational literature. We break down the exact technical boundaries of our work across our three main results:
>
> ### Novelty delineation
> **1. The TERM Analysis (Theorem 1)**
> Our transductive analysis shares a starting point with recent literature—specifically, focusing on the excess empirical loss over the second half of a symmetrized sample. However, the concentration machinery required for Massart noise diverges significantly from prior art:
> * *Prior Art:* The closest equivalences (e.g., Hanneke & Moran, 2026; Hanneke & Xu, 2025) rely heavily on variants of uniform Bernstein inequalities to control the excess empirical loss, yielding bounds O(√(VC(H)/n)), and via substantial additional machinery to implement localization, a rate o(1/√n).
> * *Our Departure:* Rather than approaching the problem via uniform Bernstein inequalities, our primary technical novelty in Theorem 1 is the derivation of a framework that bridges the non-trivial gap between the excess empirical loss and the *offset Rademacher process*. By formulating and controlling this offset process, we can inject the local metric entropy machinery to break the 1/√n barrier.
>
> **2. Sample Complexity Bounds (Theorem 2)**
> We derive these bounds by leveraging our novel in-expectation transductive bound combined with standard confidence boosting, which remains standard practice in modern literature (e.g., Alon et al., 2021).
>
> **3. Universal Learning Rates (The Pentachotomy)**
> The derivation of the universal rates involves a mix of adapted structural frameworks and entirely original constructions:
> * *Upper Bounds:* We adapt the overarching structural technique from Hanneke & Moran (2026). However, we replace their agnostic TERM in-expectation bounds with our Massart-specific bounds. Furthermore, we had to derive a dedicated structural result (**Lemma 17**) to show that a sufficiently good batch size can be found when universally learning a concept class lacking an infinite star tree. Lemma 17 is a novel and strictly necessary adaptation for the Massart setting.
> * *Lower Bounds (Theorem 5):* We emphasize that our lower bound construction is entirely original. We base our bounds on the information-theoretic framework established by Raginsky and Rakhlin (2011), adapting their techniques to our Massart-specific sequential setting to provide a standalone proof of the fundamental limits.
>
> **Action taken:** We have integrated a summarized version of this novelty delineation directly into the relevant sections.
>
> ### Main-Text Clarity and High-Level Roadmap
> We agree that the main text would benefit from a clearer structural on-ramp. We will modify the beginning of Section 3 to include the following explicit roadmap for the TERM analysis:
> 1. **Exchangeability:** We use exchangeability to show the problem reduces to establishing the concentration of the empirical error difference between the two halves of a symmetrized sample.
> 2. **Shifted Symmetrization:** Exploiting the fact that the excess error on the first half is non-positive, we introduce a shifting of the excess error and apply a symmetrization trick. For any concept in the projection, its excess error on the half-samples can be viewed as independent copies.
> 3. **Efficient Contraction (Label Dropping):** To advance to the localization of the concept class, we apply our efficient variant of the contraction lemma, allowing for a significantly streamlined proof.
> 4. **Local Metric Entropy Fixed-Points:** Finally, we invoke the localization machinery of Zhivotovskiy and Hanneke (2018) to prove that the resulting offset Rademacher process is controlled by the fixed-point of the local empirical entropy.
>
> Regarding the "steep presentation," we maintain that terms like *universal measurability*, *Gale-Stewart games*, and *pattern avoidance* are standard descriptive machinery required to validate/intuit universal learning results (e.g., Bousquet et al., 2021; Carmon et al., 2022).
>
> ### Subtleties in Excess-Error Definitions
> We appreciate this feedback, as precision here is critical for interpreting the sample complexity claims.
>
> 1. **Distinguishing Error Notions:** The ambiguity stems from omitting the explicit expression for the approximation error, er_P(H), in the original submission's preliminaries.
> 2. **Equivalence in Context of Learnability:** While the exact equivalence of these error notions depends on conditions like a finite VC dimension, this requirement does not practically narrow our results, as a finite VC dimension is strictly *necessary* for PAC learning. Therefore, in any learnable regime, they are functionally equivalent.
>
> **Action taken:** We have updated the Preliminaries section to explicitly define the approximation error er_P(H) to avoid confusion with \inf_{h \in H}er_P(h) \in. Furthermore, we have added a brief footnote regarding equivalence of the two notions.

---

### Decision · Program_Chairs · 2026-04-30

**Decision:**

Accept (regular)

**Comment:**

This submission studies binary classification under Massart noise for partial concept classes and universal learning rates, deriving sharper bounds for the TERM algorithm and establishing a pentachotomy of universal rates. The paper received four reviews with recommendations ranging from accept to weak reject. One of the reviewers (AC5T) raised substantive concerns regarding the clarity and completeness of key theoretical claims, particularly the projection step in Theorem 1 and the explicit justification for the exact 1/n lower bound. Notably, Reviewer T9Ek subsequently reviewed AC5T's comments and lowered their score, indicating that the issues identified were more substantial than initial review. Given the mixed scores and the specific technical reservations from the most critical reviewer, a weak accept is recommended and I believe that current version requires meaningful revision before publication.